# Microbial growth under drought is confined to distinct taxa and modified by potential future climate conditions

Dennis Metze [1,2] ✉, Jörg Schnecker [1], Alberto Canarini[1], Lucia Fuchslueger[1], Benjamin J. Koch [3], Bram W. Stone [4], Bruce A. Hungate [3], Bela Hausmann [5,6], Hannes Schmidt [1], Andreas Schaumberger [7], Michael Bahn [8], Christina Kaiser [1] & Andreas Richter [1,9] ✉

Climate change increases the frequency and intensity of drought events, affecting soil functions including carbon sequestration and nutrient cycling, which are driven by growing microorganisms. Yet we know little about microbial responses to drought due to methodological limitations. Here, we estimate microbial growth rates in montane grassland soils exposed to ambient conditions, drought, and potential future climate conditions (i.e., soils exposed to 6 years of elevated temperatures and elevated $CO_2$ levels). For this purpose, we combined [18]O-water vapor equilibration with quantitative stable isotope probing (termed 'vapor-qSIP') to measure taxon-specific microbial growth in dry soils. In our experiments, drought caused >90% of bacterial and archaeal taxa to stop dividing and reduced the growth rates of persisting ones. Under drought, growing taxa accounted for only 4% of the total community as compared to 35% in the controls. Drought-tolerant communities were dominated by specialized members of the Actinobacteriota, particularly the genus Streptomyces. Six years of pre-exposure to future climate conditions (3 °C warming and + 300 ppm atmospheric $CO_2$) alleviated drought effects on microbial growth, through more drought-tolerant taxa across major phyla, accounting for 9% of the total community. Our results provide insights into the response of active microbes to drought today and in a future climate, and highlight the importance of studying drought in combination with future climate conditions to capture interactive effects and improve predictions of future soil-climate feedbacks.

Droughts can have detrimental effects on ecosystems and human societies, threatening food production and forest survival[1,2]. In the last two decades, some areas of the world have experienced the driest period since the late 1500 s[3] and ongoing global warming is expected to increase the frequency, intensity, and duration of drought events even more[4]. Droughts will have strong effects on soils which not only harbor nearly 80% of terrestrial carbon stocks but are also home to about 25% of the global biodiversity[5,6]. Soil carbon fluxes and nutrient cycling are under the direct control of the growth and activity of soil microorganisms, governing, for instance, how much carbon will be lost to the atmosphere[7–9]. Shifts in microbial activity due to drought, warming, or elevated $CO_2$ concentrations may thus affect the direction and magnitude of these processes with consequences for the global climate and soil health. Low soil water content can limit microbial

activities in several ways, for instance, by reducing substrate availability due to a decreased diffusion or increasing the risk of cell dehydration[10,11]. Soil microorganisms have different strategies to cope with drought. Those include tolerance strategies, such as osmolyte production and higher investments in substrate acquisition via enzyme synthesis, or avoidance strategies, such as dormancy[12,13]. These strategies vary across groups of microorganisms and will determine their ability to remain active and growing under drought. We consider bacterial and archaeal taxa that show growth in drought-affected soils to be drought-tolerant. While it is known that microbial community composition is sensitive to soil dry-down and drought duration[14–17], it remains poorly understood which soil microorganisms are active and growing in dry soils, despite their ecological importance.

One gram of soil harbors tens of thousands of different taxa and up to billions of microbial cells of which only a small fraction (0.1–2%) is thought to be active at any given time[18]. More recent evidence, however, suggests that this fraction might be substantially larger (25–70%)[19]. Elucidating which microbial taxa are active and how much they grow is key to understanding the mechanisms that underlie shifts in organic matter fluxes, especially under climate change. Recent calculations estimate that one growing cell is as active as 1000 starving or 1,000,000 dormant cells, indicating that growing microbes account for >95% of total community respiration[20]. Under drought conditions, the number of growing cells may be lower because of reduced soil connectivity[21]. A recent conceptual model proposes that drought forces microorganisms to allocate more energy to adjust to the water deficit, including osmolyte production, or optimized substrate uptake, potentially reducing their growth[22]. Which microbes remain growing during drought might also influence carbon losses in response to rewetting such as the well-known burst in soil $CO_2$ emissions called the Birch effect[23–25].

Drought is among the main global change factors and is predicted to occur more frequently in a future climate when it will act together with elevated temperatures and higher atmospheric $CO_2$. Hence, studying drought under future climate conditions is crucial to understanding how microorganisms will function in dry soils in the decades ahead. While this will likely improve predictions of ecosystem processes such as carbon fluxes, data from multifactorial experiments remains scarce[26].

Currently, the most reliable tool to measure growth rates of microbial taxa in soil is quantitative stable isotope probing (qSIP) with $^{18}O$-labelled water[27,28]. In the presence of $^{18}O$-labeled water, $^{18}O$ is incorporated into the DNA of growing (i.e., dividing) microorganisms, which can be used to estimate growth rates of individual taxa[27]. This method has been used to study microbial growth patterns upon soil rewetting[29,30], but it could not be applied to estimate growth rates in dry soils since it relies on the addition of liquid water which can lead to growth overestimations of up to 250%[31]. Consequently, the growth rates and identities of growing taxa under drought have not been measured to date.

Here, we estimate bacterial and archaeal taxon-specific growth rates from soils exposed to ambient conditions, drought, future climate conditions (i.e., soils exposed to 6 years of elevated temperatures and elevated $CO_2$), and a combination of the latter, by combining $^{18}O$-qSIP with an approach to enrich soil water isotopically without adding liquid water[31]. This approach, called ´water-vapor equilibration qSIP" (vapor-qSIP), enabled us to identify growing bacterial and archaeal taxa during drought and explore the interaction of drought and future climate conditions. The study was undertaken at a multifactorial climate change experiment located in a montane grassland in the Austrian Alps, in which a 6-week-long summer drought was simulated under ambient and future climate conditions (i.e., elevated temperatures of + 3 °C above ambient, and elevated atmospheric $CO_2$ of + 300 ppm above ambient) and compared to respective drought-unaffected controls[32]. Our main goals were (i) to estimate the growth rates of soil bacteria and archaea under a severe summer drought and (ii) to assess how future climate conditions affect the microbial growth response to drought. We report the first analysis of the size, diversity, and composition of the growing microbial communities during drought, and how drought effects were altered and alleviated by six years of pre-exposure to future climate conditions. Using vapor-qSIP allowed us to infer growth rates of individual bacterial and archaeal taxa and to explore growth patterns at a high taxonomic resolution, ranging from phylum to genus and even ASV (amplicon sequence variant) level.

## Results

### Growing communities reveal the effects of future climate conditions, masked at the total community level

Drought and future climate conditions significantly reduced volumetric soil water contents across treatments by ~17% and ~1%, respectively (Two-way ANOVA; Drought: F = 1485, df = 1, $p < 0.001$; Climate: F = 4.2, df = 1, $p = 0.047$). We compared the effects of drought, future climate conditions, and their combination on the composition of the total and actively growing community of bacteria and archaea using principal component analysis on the absolute abundances of individual taxa (16 S rRNA gene copies per taxon) after centered log-ratio transformation. Total community composition shifted in response to drought (PERMANOVA: $R^2 = 0.13$), whereas it was unaffected by future climate conditions (Fig. 1a). Drought had an even stronger effect on the composition of the growing community (PERMANOVA: $R^2 = 0.24$) which, in contrast, was also shaped by future climate conditions (PERMANOVA: $R^2 = 0.11$) and their interaction with drought (PERMANOVA: $R^2 = 0.11$). Communities of growing bacteria and archaea were distinct in their composition across all four treatments with drought being the main driver of divergence. We found a similar pattern when using relative growth rates for PCA instead of absolute abundances, suggesting that taxon abundances reflected growth patterns (Supplementary Fig. 1). Overall, drought and future climate effects on community composition seemed to be only visible within the growing community but masked at the total community level, which only responded to drought.

### Growing communities become smaller and less diverse in drought-affected soils

Drought exhibited a strong negative impact on the growing community which seemed alleviated under future climate conditions. Drought substantially reduced the richness of growing bacterial and archaeal taxa represented as amplicon sequence variants (ASVs). The number of growing ASVS, as defined by minimum $^{18}O$ enrichment (atom percent excess (APE) $^{18}O > 5\%$), dropped on average by ~65% compared to ambient conditions, representing a loss of approximately 1000 growing ASVs (Fig. 2a, Two-way ANOVA; Drought: F = 39.1, df = 1, $p < 0.001$; Climate: F = 0.8, df = 1, $p = 0.38$; Drought x Climate: F = 4.1, df = 1, $p = 0.06$). Still, more than 500 ASVs remained growing in dry soils despite soil water contents being as low as 6–8% of soil fresh mass. We considered taxa that were able to grow in drought-affected soils to be drought-tolerant. Soils subjected to both drought and future climate conditions had 66% greater ASV richness than soils subject to drought only ($p = 0.06$). Future climate conditions alone did not affect ASV richness.

The drought effect on the size of the growing community, or the percentage of the total community that was growing, was even stronger. To estimate this, we summed the 16 S rRNA gene copies of all growing taxa and divided them by the sum of the gene copies of the total community (Fig. 2b). The total sum of 16 S rRNA gene copies did not significantly differ across treatments (Two-way ANOVA; Drought: F = 0.7, df = 1, $p = 0.39$; Climate: F = 2.6, df = 1, $p = 0.12$; Drought x Climate: F = 0.004, df = 1, $p = 0.95$). The share of growing taxa, however, declined from 35% under ambient conditions to only 4% in dry soils

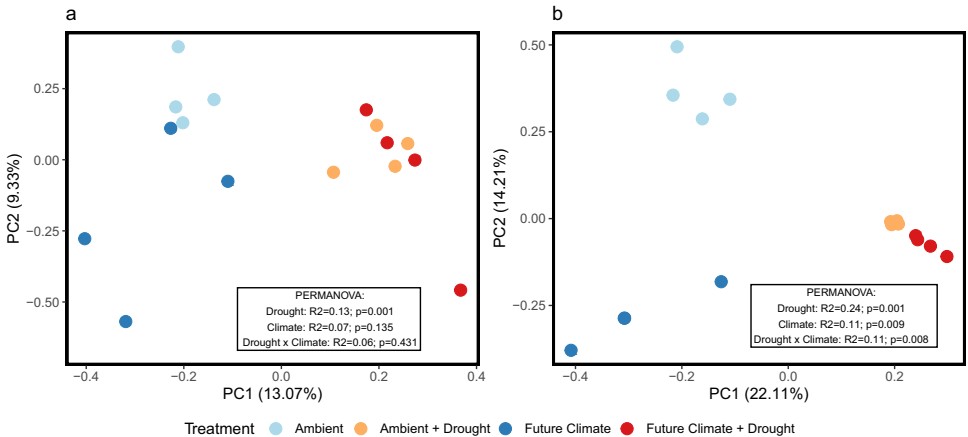

**Fig. 1 | Effect of drought and future climate conditions on the composition of total and growing bacterial and archaeal communities.** Principal component analysis (PCA) of total (**a**) and growing (**b**) bacterial and archaeal communities ($n = 4$ replicates) on centered log-ratio transformed amplicon sequence variant (ASV) absolute abundances. Absolute abundances were calculated by multiplying ASV-specific amplicon sequencing reads with 16 S rRNA gene copies inferred by digital droplet PCR. Absolute abundances were agglomerated over the density fractions of a sample gradient. Statistics from two-way permutation-based multi-variate analysis of variance (PERMANOVA) testing a full two factorial design (Drought $_{Yes}$, Drought $_{No}$, Climate $_{Ambient}$, Climate $_{Future}$) are provided as inset panels. Source data are provided with this paper.

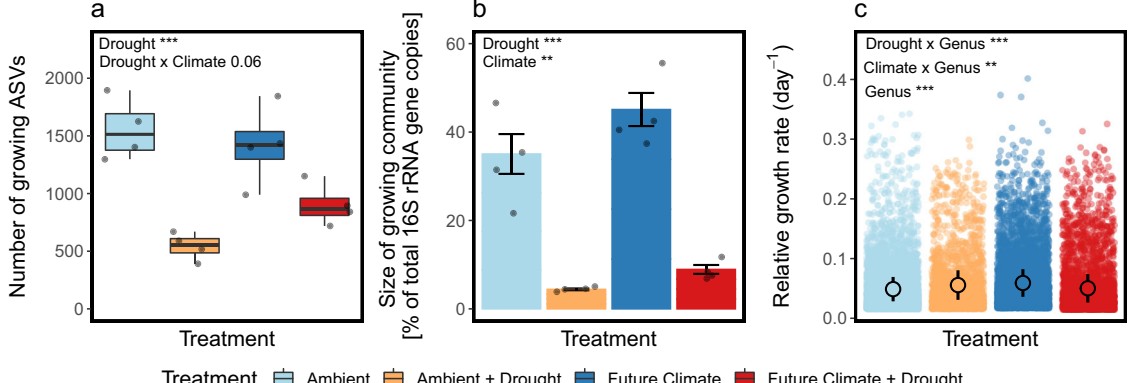

**Fig. 2 | Diversity, population size, and mean relative growth rates of growing communities of bacteria and archaea across treatments. a** Number of unique growing amplicon sequence variants (ASVs) ($n = 4$ replicates). **b** Population size of the growing community expressed as the percentage of growing taxa of the total community based on the sum of their 16 S rRNA gene copies ($n = 4$ replicates). **c** Mean taxon-level relative growth rates ($n = 4$ means of taxon-specific growth rates calculated per replicate). Asterisks depict significant results from either two-way ANOVA testing a full two factorial design (**a**, **b**: Drought $_{Yes}$, Drought $_{No}$, Climate $_{Ambient}$, Climate $_{Future}$) or a linear mixed model (**c**, $n = 391–1845$ taxon-specific growth rates per replicate). Light-colored dots represent replicate samples (**a**, **b**) or taxon-specific growth rates (**c**) across treatment replicates, used to calculate mean taxon-level relative growth rates and standard errors (points, error bars). Boxes show the interquartile range with a line representing the median and minimum/maximum whiskers (**a**). Error bars represent standard errors (**b**, **c**). Source data are provided with this paper.

(Fig. 2b, Two-way ANOVA; Drought: F = 265, df = 1, $p < 0.001$; Climate: F = 17.4, df = 1, $p = 0.001$; Drought x Climate: F = 2.4, df = 1, $p = 0.14$). Future climate conditions increased the size of the growing community compared to ambient conditions in drought–affected and drought-unaffected soils. While on average ~45% of the community was growing in future climate plots at regular precipitation, it was ~9% under drought.

### Diverging growth responses under drought and future climate conditions

We investigated how drought and future climate conditions affected bacterial and archaeal growth rates from two perspectives. First, we calculated average relative growth rates for the growing community using data from all growing taxa per treatment. Second, we assessed how relative growth rates of specific taxa changed with treatment conditions. To this end, we filtered for taxa growing in more than one treatment and compared their average relative growth rates.

Although substantially fewer taxa were growing under drought, the mean relative growth rate of the growing community did not differ from ambient conditions (Fig. 2c). We found, however, that general growth patterns and growth responses to treatment conditions varied across genera (Fig. 2c, Linear mixed model; Genus: F = 3.2, df = 376, $p < 0.001$; Climate x Genus: F = 1.3, df = 251, $p = 0.001$; Drought x Genus: F = 2.3, df = 222, $p < 0.001$). When also taking non-growing taxa into account (defined as $^{18}$O APE < 5%), representing the entire community, drought decreased mean relative growth rates while they increased in a future climate when precipitation was not manipulated (Supplementary Fig. 2, Two-way ANOVA; Drought: F = 57, df = 1, $p < 0.001$; Climate: F = 7.6, df = 1, $p = 0.01$).

To further understand how many taxa remained growing under drought and examine their growth response, we filtered for ASVs that consistently grew in at least two replicates per treatment (total growing in ≥2 replicates: 3553 ASVs, total growing: 5116 ASVs). The majority of taxa, representing about two-thirds (>2200 ASVs), were specific to

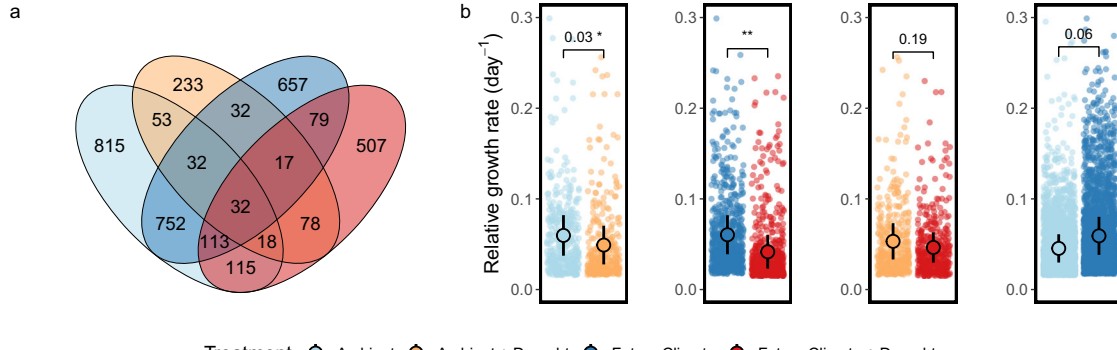

**Fig. 3 | Overlap of growing ASVs between treatments and growth response of shared taxa. a** Venn diagram of shared and unique amplicon sequence variants (ASVs) that were actively growing in at least two replicates per treatment. **b** Growth rates of shared taxa across four treatment comparisons: Ambient and Ambient + Drought (left panel), Future Climate and Future Climate + Drought (left center panel), Ambient + Drought and Future Climate + Drought (right center panel), Ambient and Future Climate (right panel). Asterisks represent significant differences between comparisons using two-sided Student's *t*-tests (*n* = 4 means of taxon-specific growth rates calculated per replicate). Light-colored dots represent taxon-specific growth rates across replicates used to calculate means and standard errors (points, error bars). Source data are provided with this paper.

only one treatment and found growing only there (Fig. 3a). In an ambient climate, only 7% of the original taxa continued to grow during drought (135 ASVs = 53 + 32 + 32 + 18 ASVs) as compared to 14% in a future climate (241 ASVs = 113 + 32 + 17 + 79 ASVs). If taxa managed to grow in both drought-affected and drought-unaffected soils, we considered them to be drought-enduring. Based on this assumption, more than twice as many ASVs were drought-enduring under simulated future climate conditions (374 ASVs = 79 + 17 + 32 + 113 + 115 + 18 ASVs) as compared to drought in an ambient climate (184 ASVs = 32 + 17 + 18 + 32 + 32 + 53 ASVs). Even though several ASVs were enduring drought conditions, their relative growth rates were reduced by 18.7% ± 5.3% (mean ± SD) in an ambient climate and by 31.2% ± 14.4% in a future climate (Fig. 3b: left panel, Ambient vs. Ambient + Drought, *t*-test: t (5.8) = 2.76, *p*-value = 0.03; Fig. 3b: left center panel, Future Climate vs. Future Climate + Drought, *t*-test: t (5.9) = 3.72, *p*-value = 0.009). At a phylum level, we observed the same response for Actinobacteriota and Bacteroidota but not Acidobacteriota and Proteobacteria (Supplementary Fig. 3A and C). If soils weren't exposed to drought, future climate conditions accelerated the growth of taxa shared with the ambient treatment by 34.7% ± 28.9% (Fig. 3b: right panel, Ambient vs. Future Climate, *t*-test: t (3.7) = −2.55, *p*-value = 0.06), which was mirrored by the Acidobacteriota, Actinobacteriota, and Bacteroidota (Supplementary Fig. 3E). This growth-promoting effect, however, was not detected in drought-affected soils (Fig. 3b: right center panel, Drought vs. Future Climate + Drought, *t*-test: t (4.9) = 1.49, *p*-value = 0.19).

### Response patterns to drought are specific to taxonomic groups and altered by future climate conditions

We examined phylum- and family-specific drought response patterns by comparing how the number of growing taxa changed across treatments (Fig. 4). During drought, the number of growing taxa within most phyla, that fulfilled normality and homoscedasticity requirements for two-way ANOVA, decreased by >50%, including Acidobacteriota, Proteobacteria, Planctomycetota, and Verrucomicrobiota (Fig. 4a). Only Actinobacteriota showed a slightly positive response to drought. Their number of active ASVs increased on average by ~7%. This response was primarily driven by five families including Streptomycetaceae, Nocardiaceae, Nocardioidaceae, Intrasporangiaceae, and Solirubrobacteraceae (Supplementary Fig. 4A). It is worthwhile mentioning that several taxa within the Proteobacteria and Acidobacteriota remained growing even at comparatively high rates, despite a generally negative drought response at the phylum level.

In the combined drought and future climate treatment, more taxa were growing across major phyla and families as compared to drought only (Fig. 4b and Supplementary Fig. 4B, respectively). For instance, the loss of Proteobacteria and Acidobacteriota was alleviated resulting in almost twice as many growing taxa. This also applied to the Actinobacteriota although less pronounced. Some phyla did not show this alleviated drought response including Crenarchaeota, Latescibacterota (Fig. 4a), and Bacteroidota (Parametric pairwise Wilcoxon tests: Drought vs. Future Climate + Drought: *p* = 0.58).

### Drought consolidates growth to fewer lineages and reduces growing bacterial predators

To examine which taxa contributed the most to the growth of the total community, we calculated their proportional $^{18}O$ assimilation (range: 0–1) by integrating relative growth rates and re-computed relative abundances for the growing community. We ranked taxa based on proportional $^{18}O$ assimilation and extracted the top five (Fig. 5) and top 50 assimilators (Supplementary Figs. 5, 6) of each sample. Across all respective samples, the top five assimilators represented 26 ASVs in drought-unaffected (Fig. 5a: ambient, future climate) and 19 ASVs in drought-affected soils (Fig. 5b: drought, future climate and drought) and accounted for 11.4 ± 4% and 23 ± 15% of the community growth (Supplementary Fig. 7A). At ambient precipitation, the top $^{18}O$ assimilating taxa included an unclassified member of the Bacteroidota (Family: env.OPS 17) and members of the genera *Pseudoduganella*, *Cavicella*, or *Bradyrhizobium* (Fig. 5a). During drought, the identity of dominant taxa shifted completely, and growth was more consolidated into fewer genera (Fig. 5b). These were mostly Actinobacteriota, belonging to the genera *Oryzihumus, Marmoricola, Rhodococcus*, and *Streptomyces*, but also included a member of the Proteobacteria. *Streptomyces* alone was responsible for on average 12.3 ± 11.1% (min: 2%, max = 38%) of the community growth in drought-affected soils (Fig. 5b). This response was primarily driven by one specific *Streptomyces* ASVs (ASV_rmw_7xq_0rh) (Supplementary Fig. 7B). This taxon maintained relative growth rates comparable to ambient conditions but increased in abundance by several magnitudes (Supplementary Fig. 8A, B).

Beyond the top $^{18}O$ assimilating taxa, we also aimed to understand the response of bacterial predators to drought. They play an important functional role in the microbial food web and might be more isolated from their prey in dry soils due to disconnected pores. Taxa were assigned a putative predatory lifestyle based on their taxonomy, for instance, if they belonged to the orders Myxococcales, Bdellovibrionales, Vampirovibrionales, Haliangiales, or Polyangiales. The vast

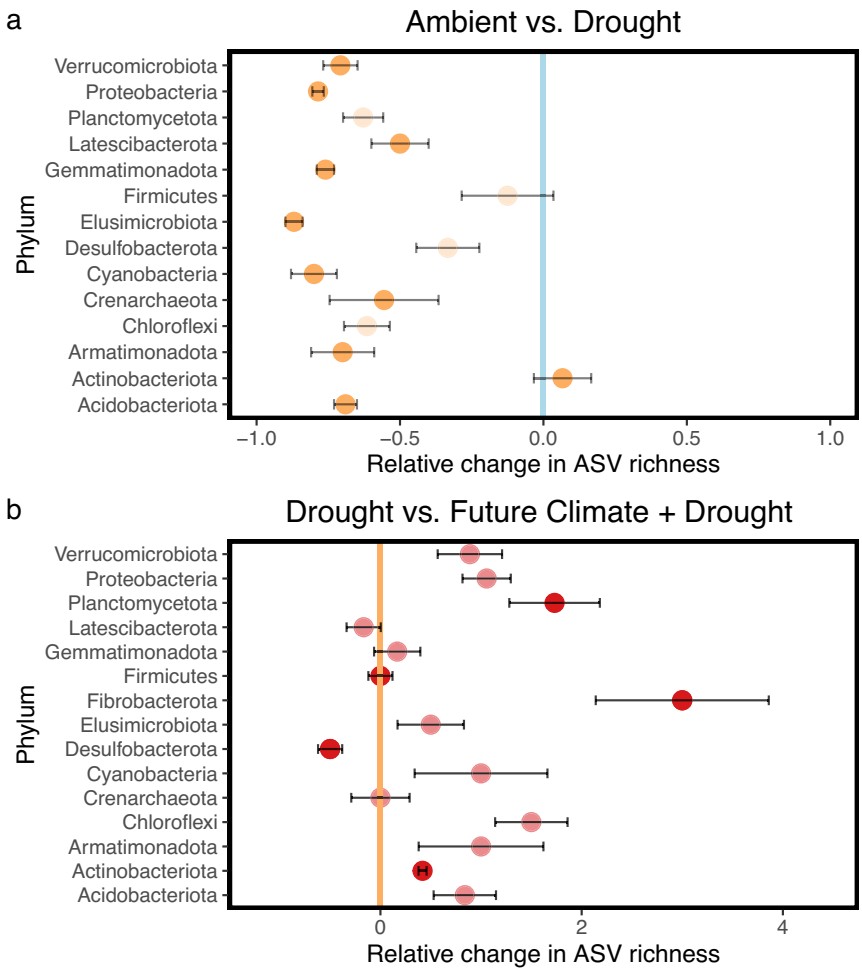

**Fig. 4 | Relative changes in the number of growing taxa per phylum under drought and future drought conditions. a** Relative change (0 = no change, 1 = increase by 100%) in the number of growing taxa per phylum comparing Ambient (reference) and Ambient + Drought conditions (*n* = 4 replicates). **b** Relative change in the number of growing taxa comparing Ambient + Drought (reference) and Future Climate + Drought conditions (*n* = 4 replicates). Points represent mean relative changes in the number of growing taxa per phylum including standard errors (error bars). Two-way ANOVA testing a full two factorial design (Drought $_{Yes}$, Drought $_{No}$, Climate $_{Ambient}$, Climate $_{Future}$) was performed on the number of growing amplicon sequence variants (ASVs) across all treatments for each phylum independently that fulfilled normality and homoscedasticity requirements. *P*-values were adjusted for multiple testing using false discovery rate correction. Bold colored points represent significant effects (**a**: Drought = $p < 0.05$, **b**: Climate = $p < 0.05$ or Drought x Climate = $p < 0.05$) and transparent points represent non-significant effects. Vertical lines depict the means of the respective reference treatment (*n* = 4 replicates) used to calculate relative changes. Source data and *p*-values are provided with this paper.

majority of these putative predatory bacteria stop growing under drought. While on average 146 ± 17 putative predatory taxa were growing at ambient precipitation, it was 15 ± 8 in drought-affected soils. With -90%, this exceeded the drought response of the total community where the richness of growing taxa averaged over both drought treatments was approximately 50% lower as compared to ambient precipitation (ambient, future climate). The contribution of putative predatory bacteria to the total community's growth based on proportional $^{18}$O assimilation also decreased with drought (Supplementary Fig. 9A, Two-way ANOVA; Drought: F = 63, df = 1, $p < 0.001$) along with a > 90% reduction in their abundance (Supplementary Fig. 9B, Two-way ANOVA; Drought: F = 227, df = 1, $p < 0.001$). Overall, most putative bacterial predators stopped growing in dry soils and even more as compared to the rest of the community. Putative predatory taxa that were still active under drought predominantly belonged to the Haliangiaceae family (Supplementary Fig. 10).

## Discussion
Drought, warming, and elevated $CO_2$ concentrations are major global change factors that alter biogeochemical processes in soils, affecting global carbon cycling and possible climate feedbacks. Growing soil microbes are at the core of these processes. While several studies have investigated the effects of drought on microbial life[10,12,16,24,33–35], our knowledge about growing microbes and their dynamic activity patterns remains limited. Furthermore, much of what we know about microorganisms in dry soils is based on total community assessments, rarely differentiating between growing (i.e., dividing), inactive, dormant, or even dead microorganisms. This might explain why results are often ambiguous with regards to shifts in the community composition and seldomly report reductions in diversity[13,36–40] or even biomass[15,41,42]. Using vapor-qSIP, we could circumvent the limitations related to liquid tracer addition, allowing new insights into microbial life in dry soils. We observed that drought caused a large part of the bacterial and archaeal community to stop growing, resulting in smaller, less diverse, and distinct active communities dominated by specific members of Actinobacteriota. In a simulated future climate, we found these drought responses to be altered and even alleviated. Thus, experiments manipulating drought as a single factor might fail to realistically capture how soil microorganisms will react to drought in a future climate.

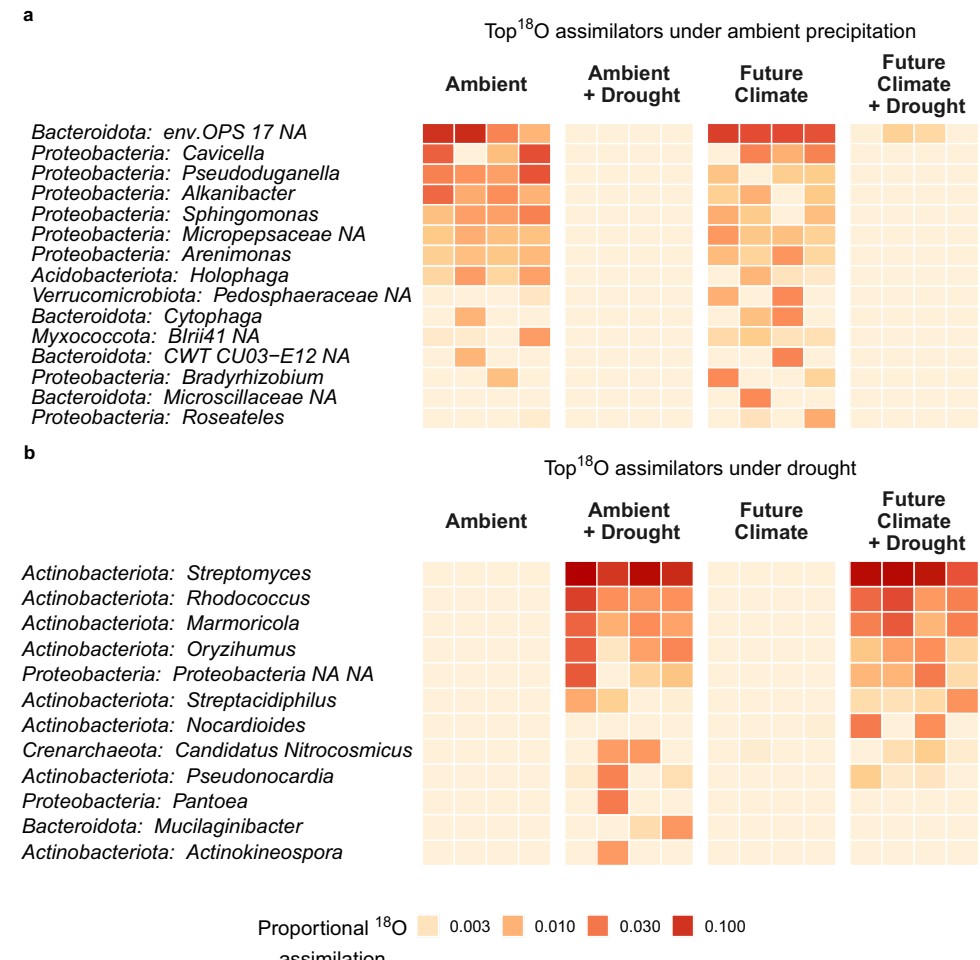

**Fig. 5 | Drought-induced changes in the top $^{18}$O assimilating taxa agglomerated at the genus level.** Heatmap showing taxa with the highest proportional $^{18}$O assimilation (contribution to the total community's growth) under ambient precipitation (**a**) and drought (**b**), visualized across all treatments and individual samples (rectangles, $n = 4$). Amplicon sequence variants (ASVs) were ranked based on their proportional $^{18}$O assimilation, separately, for drought-unaffected (**a**: Ambient, Future Climate) and drought-affected samples (**b**: Ambient + Drought, Future Climate + Drought). The top five ASVs per sample were then selected (**a**: 26 total unique ASVs; **b**: 19 total unique ASVs) and visualized. Proportional $^{18}$O

assimilation ranges from 0–1 and estimates how much a single taxon contributes to the community's overall growth. It is calculated using re-computed relative abundances of only growing taxa (sum of growing taxa = 1) and their relative growth rates (RGR). ASVs had to be active in at least two samples if detected as growing in a treatment. ASV identities were agglomerated at the genus level and sorted in descending order based on proportional $^{18}$O assimilation. If genus identity could not be assigned (NA), we agglomerated taxa at the family or phylum level. Source data are provided with this paper.

As soils dry down the water potential decreases, exposing microorganisms to an array of stressors including disconnectivity and, at local scales, resource limitation[13]. Our results suggest that most taxa active at ambient conditions are not well equipped for growth under drought conditions. Some taxa, however, were able to withstand drought and maintained growth in drought-affected soils but at on average 19% lower rates (Fig. 3b). We considered taxa to be drought-tolerant if they showed growth in drought affected-soils. We acknowledge that some taxa might have been drought-tolerant but did not grow, though these are likely to be less well-adapted. Drought adaptations include, for example, osmolyte production or biofilm formation. However, such strategies are energetically expensive[22] and reduce resources allocated to reproduction, leading to lower growth rates. Here, Actinobacteriota were the most drought-tolerant phylum with even slightly increasing numbers of growing taxa (Fig. 4a). This underlines the persistence of Actinobacteriota in dry soils previously indicated by higher relative abundances[17,24,43,44]. Extending such findings here, the positive response of Actinobacteriota was driven by a few families including the Streptomycetaceae, Nocardiaceae, Nocardioidaceae, and Intrasporangiaceae (Supplementary Fig. 4), and only a

few genera within them, namely *Oryzihumus, Rhodococcus, Marmoricola*, and *Streptomyces*. Among these, *Streptomyces* accounted for most of the community's growth, underlining its paramount role under drought. A similar pattern of growth consolidation has been seen upon nutrient addition[45]. *Streptomyces* maintained the same growth rates as under ambient conditions while increasing in abundance. Their filamentous growth might be advantageous when pore spaces and resources become disconnected[21]. *Streptomyces* has been observed to persist in dry soils, alleviating drought stress in plants[44,46–52]. Interestingly, individual taxa from mostly drought-sensitive phyla also grew in dry soils such as one taxon assigned as Proteobacterium (Fig. 5b), despite more than 70% of the Proteobacteria becoming inactive. This indicates that although certain traits related to drought tolerance are phylogenetically conserved, others might be more widespread and even found in taxa of generally drought-sensitive groups where they might have evolved independently or were acquired via horizontal gene transfer[53,54].

Since pore spaces become increasingly disconnected with drought, leading to lower diffusion and mobility, microbe-feeding predators are expected to have less access to their prey, potentially

decreasing predatory pressure[13,21]. While this has been demonstrated for soil protists and nematodes before[55,56], we demonstrate here that most predatory bacteria almost completely ceased their growth, too. Although we could not differentiate between facultative and obligate predatory taxa, our results suggest a strong decrease in predation via predatory bacteria. Our growth data does not explain the reasons behind this decrease, but the disproportionate reduction of growing predatory bacteria as compared to the community's average might indicate that drought entails additional challenges for predatory bacteria such as restricted predator movement and lower prey accessibility due to drought.

In our study, microbial drought responses changed when soils had also been pre-exposed to future climate conditions (+3 °C and +300 ppm). Generally, future climate conditions allowed for larger active communities (Fig. 2b) and, at ambient precipitation, also higher relative growth rates (Fig. 3b: right panel). In drought-affected soils, future climate conditions alleviated the loss of growing taxa across many phyla. Furthermore, growing taxa under combined conditions formed distinct communities including many unique ASVs.

We hypothesize that the attenuated drought response of the growing community under future climate conditions might arise from (I) the 6-year-long pre-exposure of the microorganisms to higher temperatures and/or (II) larger carbon inputs from plants. It is well-known that previous exposure to drought often increases microbial resistance and resilience[17,57,58], but to our best knowledge, this is the first report that future climate conditions can alleviate negative drought effects on soil microorganisms. Compositional changes and associated shifts in the distribution of life-history traits in response to drying and rewetting can render microbial communities more drought-tolerant[54]. The simulation of future climate conditions significantly reduced the soil moisture content during the drought experiment and in the past, where they led to an earlier and more frequent drop in soil moisture[59,60], exposing the microbial community to longer and more frequent dry periods. This has likely caused a selection for more drought-enduring taxa and populations.

With regard to plants, higher concentrations of atmospheric $CO_2$ can increase leaf and root production[61] as well as exudation[62]. Higher carbon allocation belowground might explain why future climate conditions led to a larger growing community and higher growth rates at ambient precipitation. Drought generally reduces plant carbon transfer to soil bacteria[15] whereas moderate drought has been shown to increase rhizodeposition[63]. Furthermore, drought can affect root exudate properties, leading to higher microbial respiration compared to ambient soils[64]. We hypothesize that increased rhizodeposition, either preceding or in an early stage of the dry phase, partially reduced microbial drought stress by alleviating resource limitations[13]. In addition, elevated atmospheric $CO_2$ might have reduced plant water uptake due to lower stomatal conductance and transpiration[65], allowing more water to remain in the soil for longer, although we found only small differences in soil water content between ambient and future climate treatments.

Growing microbial taxa are key contributors to soil carbon cycling. They transform organic carbon to produce new biomass and release $CO_2$ produced for energy production back to the atmosphere, more than dormant and starving taxa combined[20]. Microbial growth rates inferred by $^{18}O$-qSIP were found to be directly related to ecosystem-level respiration rates in soil[45]. Here, the growing community changed in both composition and size under drought and future climate conditions as well as their combination, which might explain previous results, reporting lower soil respiration under drought and slightly higher respiration when previously exposed to future climate conditions[66].

Our results illustrate that vapor-qSIP, i.e., the combination of $^{18}O$ water vapor equilibration and qSIP, allows for unprecedented insights into growing microbial communities in dry soils, revealing that future

climate conditions increased their drought tolerance. Understanding which microorganisms persist to grow in dry soils and at what rates constitutes the basis of an in-depth understanding of how drought affects soil carbon cycling and soil organic matter persistence. Considering that microbes can alleviate drought stress in plants, understanding which taxa grow in dry soils, can be exploited to engineer rhizosphere microbiomes for future climatic conditions. While members of the genus *Streptomyces* are well known for reducing drought stress in plants, we present other drought-tolerant genera such as *Oryzihumus, Rhodococcus,* and *Marmoricola* which might prove interesting for future studies. With regards to soil functioning, the vast majority of taxa active at ambient conditions stopped growing under drought and were replaced by a substantially smaller number of drought-tolerant taxa. This will reduce the rate of many biogeochemical processes as well as the diversity of the active microbiome, rendering it more vulnerable to additional disturbances, such as environmental contamination. While this provides new directions for drought research, we highlight that it is important to study drought in combination with future climate conditions to capture interactive effects and improve predictions of future soil-climate feedbacks.

## Methods

### Study site and sample collection

This study was conducted within the ClimGrass experiment located at the Agricultural Research and Education Centre, Raumberg-Gumpenstein, Austria (47°29′38″N, 14°06′03″E, 710 m, MAT: 8.2 °C, MAT: 1056 mm). ClimGrass is a multifactorial climate change experiment established in a managed grassland and fully operational since 2014. It comprises 54 experimental plots and six treatment conditions characterized by joint or individual manipulations of temperature (ambient, +1.5 °C, +3 °C), atmospheric $CO_2$ (ambient, +150 ppm, +300 ppm), and severe summer drought events[32,66–68]. Soil temperatures and $CO_2$ concentrations were manipulated with infrared heaters and miniFACE systems since 2014 (Free Air $CO_2$ enrichment System), respectively. To simulate a summer drought event, automated rainout shelters were installed for six weeks (June 17th–August 3rd, 2020). The soil type was a Cambisol (pH ~5.5) with a loamy sand texture. Soil carbon and nitrogen content were $3.2 \pm 0.3\%$ (mean ± SD) and $0.3 \pm 0.04\%$ per gram of dry soil, respectively. Dominant plant species included the grasses *Arrhenatherum elatius* and *Festuca pratensis,* the legumes *Lotus corniculatus* and *Trifolium pratense,* and the non-leguminous forbs *Taraxacum officinale* and *Plantago lanceolata*[67].

At the end of July (July 29th, 2020), soil samples were collected from the top 10 cm with a corer (2 cm diameter) from four treatments and all four respective replicates plots each: (i) ambient conditions, (ii) ambient conditions & drought, (iii) future climate conditions (+3 °C and +300 ppm), and (iv) future climate conditions & drought. This represented peak drought for the plots where precipitation was manipulated using rainout shelters (ambient conditions & drought, future climate conditions & drought). At the time of sampling, all future climate plots have been exposed to warming and elevated atmospheric $CO_2$ for ~6 years. All drought plots experienced two previous summer drought simulations (2017, 2019).

### Quantitative stable isotope probing via $^{18}O$ water vapor equilibration (vapor-qSIP)

Soils were passed through a 2 mm sieve to remove rocks, plant litter, and roots and stored for 48 h until the start of qSIP incubations. Soil water content was determined gravimetrically by drying 2 g of fresh soil at 105 °C for 24 h.

Samples for qSIP were incubated with $^{18}O$-enriched water and water at natural abundance isotope concentrations using the $^{18}O$ water vapor equilibration method[31]. Typically, in qSIP experiments, soil samples are air-dried followed by the direct addition of a liquid tracer (e.g., $^{18}O$ water). While this is necessary to study, for instance, the

effects of rewetting on the physiology of soil microorganisms[30], this procedure changes the soil water content. Especially in dry soils, liquid water addition stimulates microbial respiration ("birch effect", Birch 1958) and growth[31,69], making it hard to derive ecologically relevant rates of microbial activity under dry conditions. In vivo [18]O water vapor equilibration avoids changes in soil water content by letting labeled water redistribute into the soil pore space from a spatially separated source in a closed system. To this end, sieved soil samples (-500 mg) were weighed in 1.2 ml cryovials and inserted into 27 ml glass headspace vials. The amount of water applied to the bottom of a headspace vial (here: 270–430 µl) was calculated using the following equation:

$$V_{18_{O-H_2O \text{ to be added}}} = \frac{V_{\text{soil water}} * {}^{18}O \text{ at}\%_{NA} - V_{\text{soil water}} * {}^{18}O \text{ at}\%_{\text{target}}}{{}^{18}O \text{ at}\%_{\text{target}} - {}^{18}O \text{ at}\%_{\text{added}}} \quad (1)$$

where $^{18}O$ at%$_{\text{target}}$ and $^{18}O$ at%$_{\text{added}}$ represent the target enrichment of the soil water (here: -70 $^{18}O$ atom%) and the enrichment of the added water, while $V_{\text{soil water}} * {}^{18}O$ at%$_{NA}$ represents the soil water volume multiplied by its natural abundance of $^{18}O$ ($^{18}O$ at%$_{NA}$ = 0.2%). The volumetric water contents of our soils were $31.6 \pm 1.8\%$ under ambient precipitation and $6.9 \pm 1.9\%$ under drought. Drought-unaffected samples were incubated with 95 atom % $^{18}O$ labeled water and drought-affected samples were incubated with 75 atom % $^{18}O$ to make sure that enough remaining water could be collected after incubation for isotopic analysis.

After water addition, we closed the headspace vials air-tight with rubber septa and incubated them at field temperatures at the time of harvest (ambient = 20 °C, future climate = 23 °C) for five days. To monitor the $^{18}O$ soil water enrichment over time, we prepared four additional $^{18}O$ calibration samples using one soil sample per treatment. From the calibration samples, we collected the remaining water at the bottom of the headspace vial after 3, 6, 24, and 48 h, respectively. At the end of the 5-day incubation period, we performed the same for all primary $^{18}O$ labeled samples. The water samples were analyzed for their $^{18}O$ enrichment through equilibration of $^{18}O$ in H$_2$O with CO$_2$ on a Gasbench II headspace sampler connected to a Delta V Advantage isotope ratio mass spectrometer (Thermo Fisher). These values informed the mean $^{18}O$ enrichments of soil water over the course of the incubation by fitting a negative exponential function and determining its integral[31]. Based on these calculations, our samples reached the target enrichment after 24–48 h (Supplementary Fig. 11), resulting in an average soil water enrichment of $59.3 \pm 3$ atom% (mean ± SD). Soil water enrichments ranged between 55–64 atom% and we accounted for this variation while calculating relative growth rates.

After incubation, soil aliquots were flash-frozen and stored at −80 °C. We isolated DNA from soils using the FastDNA™ SPIN Kit for Soil (MO Biomedicals) following manufacturer's instructions. DNA concentrations were quantified fluorometrically using the PicoGreen assay (Quant-iT™ PicoGreen dsDNA Reagent, Life Technologies). To measure $^{18}O$ isotope incorporation in microbial DNA, we subjected our samples to ultracentrifugation in a cesium chloride density gradient[27]. We loaded 2 µg DNA into a 4.7 mL OptiSeal ultracentrifuge tube (Beckman Coulter) with ~4 ml saturated CsCl solution and gradient buffer (100 mM Tris, 100 mM KCl, 1 mM EDTA). Samples were spun in a Beckman Optima ultracentrifuge using a Beckman VTi 65.2 rotor (50,000 rpm at 20 °C) for 72 h. We manually collected 24 fractions of 250 µl after puncturing the tubes with a cannula (Braun Sterican, 0.9 × 25 mm). Sterile distilled water served as a displacement medium. During fractionation, tubes were secured with a three-prong clamp attached to a retort stand. The density of each fraction was determined with a Krüss DR301-95 digital refractometer. DNA was purified from the CsCl solution by glycogen-aided isopropanol precipitation, resuspended in 50 µl of nuclease-free water, and quantified by the PicoGreen assay (see above).

Sequencing was performed on 15-16 fractions per sample at the Joint Microbiome Facility of the Medical University of Vienna and the University of Vienna (JMF project IDs: JMF-2104-06, JMF-2012-8). Fractions were selected based on DNA content and density, excluding those without detectable DNA and mostly consisting of displacement medium. A two-step barcoding approach was used to generate amplicon libraries of archaeal and bacterial communities using Illumina MiSeq (V3 Kit) in the 2 × 300 bp configuration[70]. Primer sequences (515F–806 R) and PCR amplification protocols (30 cycles) were used as specified by the Earth Microbiome Project[71] standard protocols (https://earthmicrobiome.org/protocols-and-standards/16s/). We used the following cycling conditions: initial denaturation at 94 °C for 4 min, 7 cycles of 94 °C for 30 s, 52 °C for 30 s, 72 °C for 60 s, and final elongation at 72 °C for 7 min[70]. Amplicon pools were extracted from the raw sequencing data using the FASTQ workflow in BaseSpace (Illumina) with default parameters. Demultiplexing was performed with the python package demultiplex (Laros JFJ, github.com/jfjlaros/demultiplex) allowing one mismatch for barcodes and two mismatches for linkers and primers[70]. Amplicon sequence variants (ASVs) were inferred using the DADA2 R package[72] applying the recommended workflow. FASTQ reads 1 and 2 were trimmed at 220 nt and 150 nt with allowed expected errors of 2 and 2, respectively. ASV sequences were subsequently classified using DADA2 and the SILVA database SSU Ref NR 99 release 138.1[73,74] with a confidence threshold of 0.5. Datasets were deposited in the NCBI Sequence Read Archive under BioProject accession number PRJNA937073.

We measured the concentration of archaeal and bacterial 16 S rRNA gene copies per fraction with the Bio-Rad QX200 Droplet Digital PCR (ddPCR) system using the same primers as for the sequencing. Individual PCR reactions comprised the following components: 11 µl 1× EvaGreen Droplet Generation Mix (Bio-Rad), 0.2 µl forward primer (10 µM), 0.2 µl reverse primer (10 µM), 8.6 µl nuclease-free water, and 2 µl diluted DNA template. Prior to ddPCR quantification, DNA samples were diluted to 0.05 ng/µl as 0.1 ng of total DNA per reaction was found optimal for the separation of negative and positive droplets. The following cycling conditions were used for ddPCR: 95 °C for 5 min, 5 cycles of 95 °C for 30 s, 57 °C for 2.5 min (−1 °C each step), followed by 35 cycles of 95 °C for 30 s, 52 °C for 2.5 min, followed by 4 °C for 5 min, and 90 °C for 5 min. Droplets were stored at 4 °C for a least one hour before reading. We used QuantaSoft software (Bio-Rad) to calculate 16 S rRNA gene copies.

## qSIP and statistical analyses

All analyses were performed in R 4.1.1[75]. Amplicon sequencing data was manipulated using the phyloseq package[76]. We removed ASVs without taxonomic assignment at the phylum level as well as sequences classified as eukaryotes, mitochondria, or chloroplasts (2243 ASVs). In addition, we removed contaminant ASVs identified by decontam 1.6.0[77] using the prevalence method and a threshold setting of 0.01 (number of negative controls = 3). Negative controls from DNA extraction and dilution steps served as input data, resulting in the removal of 11 ASVs. Only samples within a density range of 1.614- – 1.753 and >2000 read pairs were retained for further analyses to exclude fractions contaminated with the fractionation medium (water) and of low sequencing yield. After filtering, we continued with 12-14 fractions per sample. Several low-density fractions of two replicates from the control treatment (ambient conditions) were lost during fractionation, resulting in higher weighted average density (WAD) values per tube. To account for this, we calculated the mean offset in weighted average density at the tube level and subtracted it from the measured densities of these samples. These procedures yielded a feature table with 15,565 ASVs and 4,342,798 read pairs.

Taxon-specific $^{18}O$ atom percent enrichment (APE $^{18}O$), a proxy for relative microbial growth, was determined based on the relationships between $^{18}O$ incorporation, DNA density, GC-content, and DNA

molecular weight[27]. Although growth and activity do not always correspond, since non-growing microbes can still show low activity, we used both terms interchangeably for simplicity reasons. We used publicly available code to perform qSIP calculations (https://bitbucket.org/QuantitativeSIP/qsip_repo, https://github.com/bramstone/qsip). A prevalence filtering step was applied prior to qSIP analysis to exclude infrequent taxa[45]. Taxa had to occur in at least 4 fractions and two replicates per treatment, yielding 6,054 ASVs accounting for 84.9 % of all sequence read pairs. We calculated replicate-level APE $^{18}$O values without bootstrapping, yielding 5,652 active ASVs (APE $^{18}$O > 0; min. APE $^{18}$O = 0; max. APE $^{18}$O = 110%). We then applied a minimum enrichment threshold (APE $^{18}$O > 5%) to ensure enrichment was due to isotopic incorporation and not caused by density variations between tubes, resulting 5116 ASVs. Taxon-level relative growth rates per day (RGR) were calculated as follows assuming linear growth: RGR = APE $^{18}$O$_{taxon}$ / (Average APE $^{18}$O$_{soil\ water}$ * 5 days)[78,79]. For this, APE $^{18}$O percent values were converted into decimals also called $^{18}$O atom fraction excess (AFE $^{18}$O) and commonly used in qSIP studies. We compared RGRs of shared taxa between treatments only if they were growing in at least two replicates each. To avoid pseudoreplication (multiple taxon-specific growth estimates per sample), we calculated mean relative growth rates for each sample and compared them using either two-way ANOVA (see details below) or Student's $t$-test for pairwise treatment comparisons.

Absolute abundances (16 S rRNA gene copies per ASV) were calculated by multiplying their sequencing-inferred relative abundances by the total number of 16 S rRNA gene copies accumulated over all density fractions of a sample. To determine the size, or percentage, of the growing community, we summed the absolute abundances of all growing taxa before dividing them by the total number of absolute abundances per sample.

Although some taxa might be growing faster than others, their contribution to the overall community-level growth will also depend on their abundance. Therefore, we estimated proportional $^{18}$O-assimilation for each growing ASV by re-calculating their relative abundances (sum of relative abundances of growing taxa = 1) and multiplying them by their relative growth rate (RGR). These weighted enrichment values were then divided by their sum, producing a proportional $^{18}$O-assimilation value ranging between 0–1. We ranked ASVs that were active in at least two replicates based on their proportional $^{18}$O-assimilation to identify the top five and top 50 $^{18}$O assimilating taxa per sample. The pool of the top five $^{18}$O assimilating taxa consisted of 26 and 19 ASVs for samples exposed to ambient precipitation and drought, respectively. We visualized changes in proportional $^{18}$O-assimilation with heatmaps using functions from the ampvis2 package[80]. For heatmaps, ASVs were agglomerated at the genus level. If genus identity could not be assigned (´NA"), we agglomerated taxa at the family or phylum level.

To compare the effect of drought and future climate treatments across phyla and families, we calculated their number of growing taxa in each sample. These numbers were used to perform two-way ANOVAs testing a full two factorial design (Drought $_{Yes}$, Drought $_{No}$, Climate $_{Ambient}$, Climate $_{Future}$) for each phylum and family individually if normality (Shapiro–Wilk test) and homoscedasticity (Levene's test) requirements were fulfilled. If these requirements were not met for a phylum, we used pairwise Wilcoxon signed-rank tests. Shifts in the number of growing taxa were visualized between treatment pairs using relative changes as compared to the mean of the respective reference (0 = no change, 1 = increase by 100%).

We compared treatment effects on the composition of total versus growing communities by computing Euclidean distances according to a new quantitative sequencing framework used when dealing with absolute abundances[81]. This was followed by two-way PERMANOVA testing a full two factorial design (Drought $_{Yes}$, Drought $_{No}$, Climate $_{Ambient}$, Climate $_{Future}$) using the ´adonis" function[82]. For the

total and growing communities, we merged the absolute abundances of all density fractions per sample (only labeled) before centered log-ratio transformation. For the growing community, we additionally computed Euclidean distances based on relative growth rates to examine differences in activity patterns beyond abundance shift. Euclidean distances were visualized using PCA.

Due to the full two factorial design of our experiment, we used two-way ANOVA to test for the effects of drought (Drought $_{Yes}$, Drought $_{No}$), climate (Climate $_{Ambient}$, Climate $_{Future}$), and their interaction on the richness, size, and mean relative growth rates (including ´non-growing" taxa) of the growing community. We also used two-way ANOVA to test for drought and climate effects on putative predators and the proportional $^{18}$O-assimilation across the top five $^{18}$O assimilating taxa including *Streptomyces* as well as on its abundance and RGR.

We examined the normality and homoscedasticity of our data and applied log transformation if necessary. False discovery rate (FDR) correction was employed to correct for multiple testing. To examine the impact of genus identity on mean relative growth rates (excluding ´non-growing" taxa) in addition to drought and climate, we used a hierarchical linear mixed model implemented in the 'lme4' package[83]. Our model had the following structure: lmer (formula = log (Relative growth rate$_{ASV}$) ~ Drought * Climate * Genus + (1|SampleID), data = data). By implementing 'Genus' in the fixed and 'Sample ID' in the random terms, we tried to limit pseudoreplication.

### Reporting summary
Further information on research design is available in the Nature Portfolio Reporting Summary linked to this article.

## Data availability
DNA sequence data generated in the frame of this study have been deposited in the NCBI Short-Read Archive under the BioProject accession number PRJNA937073. The data used in this study are available in the Zenodo database under accession code 8109566. Source data are provided with this paper.

## Code availability
All code and data used to produce the analyses and figures of this study are openly available in the Zenodo database under accession code 8109566.

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

## Acknowledgements

DM was financially supported by FutureArctic, a European Union's Horizon 2020 research and innovation program under the Marie Skłodowska-Curie Actions (grant no. 813114). We thank Jasmin Schwarz and Gudrun Kohl for 16 S rRNA gene amplification and sequencing. We also thank Joana Senéca Silva for handling the submission of the sequencing data to the NCBI Short-Read Archive, Petra Pjevac for her helpful comments on the manuscript, and Leila Hadziabdic for optimizing the conditions for 16 S rRNA gene amplification using ddPCR. Furthermore, we thank Stephanie Eichorst for her support with the ultracentrifuge.

## Author contributions

AR, DM, JS, LF, AC, and CK conceived the study, and AS, MB, and AR established the field site. AC, LF, and JS carried out fieldwork and incubations. DM performed the lab experiments and data analyses, with important inputs from BWS, BJK, HS, and BAH. BH processed the raw amplicon sequencing data.

## Competing interests

The authors declare no competing interests.

## Additional information

[1]Centre for Microbiology and Environmental Systems Science, University of Vienna, Vienna, Austria. [2]Doctoral School in Microbiology and Environmental Science, University of Vienna, Vienna, Austria. [3]Center for Ecosystem Science and Society and Department of Biological Sciences, Northern Arizona University, Flagstaff, AZ, USA. [4]Earth and Biological Sciences Directorate, Pacific Northwest National Laboratory, Richland, WA, USA. [5]Joint Microbiome Facility of the Medical University of Vienna and the University of Vienna, Vienna, Austria. [6]Division of Clinical Microbiology, Department of Laboratory Medicine, Medical University of Vienna, Vienna, Austria. [7]Agricultural Research and Education Centre Raumberg-Gumpenstein, Irdning, Austria. [8]Department of Ecology, University of Innsbruck, Innsbruck, Austria. [9]International Institute for Applied Systems Analysis, Advancing Systems Analysis Program, Laxenburg, Austria. ✉e-mail: dennis.metze@univie.ac.at; andreas.richter@univie.ac.at

