## [Peer Review File · Nature Communications]

REVIEWER COMMENTS

Reviewer #1 (Remarks to the Author):

NCOMMS-23-08901

The manuscript by Metze et al. examined the growth response of soil bacteria to a combination of six-week drought (i.e., drought manipulation) and 6-year elevated temperature and CO₂ manipulations (i.e., "future climate" manipulation). The authors found that previous exposure to future climate conditions attenuated the drought response. That is, while drought reduced the number of growing taxa by about 2/3 relative to ambient conditions, the number of growing taxa was only reduced by about 1/3 when the community was previously exposed to future climate with subsequent drought manipulation (Fig 2A). Further, the authors focused on growing taxa using 18-O qSIP modified with a previously described vapor equilibration technique that allows measurement of the growth response of microbes in dry soils without adding excess water (and thus affecting the growth response). While I am not an expert on soil water physics, and thus can not comment on the specifics of the vapor equilibration method, the technique certainly could provide an advance in understanding of microbial growth in dry soils. Given this, the authors show that while no changes in community composition due to drought were detected in their experiment using DNA alone, they did detect changes in community composition using the 18-O labeling vapor equilibration technique, pointing to a subset of taxa that were able to maintain growth even under a severe drought (Fig 1).

In general I find the paper to be well-written, however there are some points where I feel either clarification is needed or the presented data do not support the claims. For example, the authors discuss "drought resistant" taxa prominently in the abstract (L44) and in the discussion (L222 and L267), however there was no formal analysis or definition of what the authors consider "drought resistant" taxa. While this is partly a matter of semantics, it is impossible for the reader to evaluate statements such as "even for drought-resistant taxa, growth was often compromised" (L222) without knowing which taxa the statement applies to. There are data presented that could potentially be used to classify drought resistance, such as the taxa that maintain high growth under drought (Fig 5B) or the taxa that were unique to the drought treatment (Fig 3A Venn diagram), but I feel a further analysis and clear explanation of what the authors consider to qualify as drought resistance is needed to support this. E.g., how were the "top 18-O assimilators" in Fig 5B selected?

In addition there are some claims that do not appear to be supported by the data as presented, specifically, 1) at L148-150 I can not sum the subsections of the Venn diagram (Fig 3A) to find the figures the authors describe and so I question whether the numbers should be rechecked or the wording clarified, 2) the sections on bacterial predators at L197-201 (results) and L249 (discussion) does not appear to be supported by any statistics or in-depth analyses, and so it is hard to find much support for

the conclusions drawn. Finally, the authors discuss the attenuation of drought response in the context of future climate as a possible response to reduced moisture levels in the heated treatments (L263-267). I agree that this is a distinct possibility. However, the authors do not prominently present any data on soil moisture, except for as an aside at a different point in the discussion (L278-279), or discuss any previously published soil moisture data from the experiment. I suggest featuring a discussion about any available data on soil moisture as this will lend credence to the conclusions drawn from the current data.

I would also suggest at least one modification to one of the primary statistical methods: the authors computed community dissimilarity based on Euclidean distance (Fig 1). However, this distance metric is widely considered to be inappropriate for ecological community composition analyses (e.g., McCune and Grace, *Analysis of Ecological Communities*, 2002). Unless the authors can justify this choice I would suggest re-analyzing the data using an appropriate distance metric.

More details on these comments and additional comments are given below.

Detailed comments:

L412 I suggest relating this back to the 15-16 density fractions per sample cited above at L375. I.e., how many fractions result from filtering to the 1.614-1.753 density range? You might also consider giving some justification for why this filtering was performed for folks who are not familiar with qSIP protocols.

L426 It seems the sample/fraction number cutoff must be an integer. I suggest giving the minimum cutoff instead of "at least 25% of all fractions (~4 fractions)".

L429 I suggest providing here the range of APE 18-O and the number of taxa detected with less or greater than the cutoff given the prior filtering steps.

L457 This is unclear. What was the response matrix for calculating Euclidean distance? Growth rates? Presence-absence in either the growing or nongrowing matrix? Further, Euclidean distance is typically considered unsuitable for community data due to a high proportion of absences (and the assumption that zeros can be either real or present but undetected with no way to differentiate these two scenarios). Depending on what the input data was I suggest either adding some text to justify the use of Euc distance or recalculating with Bray-Curtis distance or another distance metric that is suitable for community data. This affects both PERMANOVA statistics and the NMDS.

L469 What is meant by "top 18-O assimilating taxa"? Is this referring to the filtering previously described? Please clarify how these taxa were selected and/or what was the cutoff.

L100-110 As noted above, Euclidean distance does not appear to be appropriate for this data set. While it was not entirely clear in the methods, based on the description of the results it seems this is community composition. I suggest recalculating with an appropriate distance metric. Further, there is really no need to use NMDS with Euclidean distance, just use PCA which is easier to interpret (axis eigenvalues).

L106-110 The effects on active community composition could be discussed with more nuance. The PERMANOVA statistics in Supp. Fig 1 clearly show that drought had by far the greatest impact on activity community 16S abundance (3x higher R^2), but also that the other effects were marginal ($P > 0.05$). In Fig 1B twice as much variation is explained by drought as climate or their interaction.

L117 Picky, but I suggest separating the statistics for the different model terms with a semicolon instead of a comma. It is easy to get lost in the long string of model terms. E.g., Drought: model coefficients, etc; Climate: model coefficients). This goes for all of such reported statistics.

Figure 2. Because you are performing mean comparisons I think the error bars in panels B and C should be changed to standard errors (an estimate of the accuracy of the sample mean relative to the population mean) as opposed to std deviation (variability of the sample).

L148-150 I think this section should either be clarified or the numbers rechecked. I can not come up with the same numbers as reported based on my understanding of Fig 3A.

- E.g., "drought only" (L148) is reported as 184 ASVs, however I interpret this as ASVs only found in drought treatment which is shown as 233 in the figure.

- "Twice as many taxa were shared between the future climate and drought treatment (374 ASVs)" I interpret as the overlap between

"Ambient + drought" and "Future climate" treatments, however these sum to 113 ASVs.

- Alternatively the whole sentence "twice as many taxa were shared between the future climate and drought treatment (374 ASVs) and the ones exposed to future climate and ambient precipitation." could also be interpreted as the overlap between "future climate + drought" and "future climate" treatments in the figure although I would then expect the number of ASVs shared to be reported at the end of the sentence. In any case the overlap between the FC and FC + drought treatment sums to 241 ASVs instead of 374.

- Further, partly because the comparisons being made are unclear, it is also unclear to me why these particular comparisons are of interest. It would help to provide a sentence outlining the rationale behind the comparisons being made, e.g., "we specifically compared overlap in growing taxa in the drought + ambient climate treatment to the future climate treatment and the FC + drought treatment to discern which taxa were responding to different pressures...."

L153 Why does it seem like some statistics are missing here? The parentheses end with calling out a panel but no stats are given.

Fig 4. I suggest modifying the x-axis label to read "Relative change in ASV richness"

Fig 5. You might consider adding an indication of the phylum to the figure as phylum is discussed prominently in the text. Also, how were "top 18-O assimilators" defined for this figure? E.g., one panel has fifteen rows by my count and the other has twelve. Given that "ASV identities were agglomerated" at the genus, family, or phylum level, are we to understand that these designations have not only variable numbers of ASVs in them, but also could represent very different levels of taxonomic cohesion? I think a better description of data treatment for this figure would be helpful.

Further, I would suggest presenting all of the data behind this figure, either at the agglomerated taxonomic level, the ASV level, or both, in a heatmap(s) in the supplement (i.e., include all taxa). You could highlight the subsections of data you are pulling out for the main text. That way the reader can make comparisons across the entire dataset if desired instead of focusing only on the particular taxa the authors choose to highlight.

L197-201 This bit about bacterial predators does not have much (any?) lead-in. Could you provide a small bit of background and/or some methodological info to describe how this was done? Further, I would like to see some statistics here instead of just a heatmap as there is a paragraph in the discussion section on these responses.

L211 I think a better job could be done with appropriate citations here. E.g., the Malik paper performed mRNA sequencing and so specifically targeted expressed transcripts and not DNA (and therefore does not support the preceding statement at L210). The Evans paper examined the effects of temporal rainfall intensity (i.e., fewer, larger rainfall events vs more frequent, smaller events), and not drought, per se, and further primarily focused on an in vitro wet up experiment over the course of a 115 day time series — where we can reasonably assume that changes in community composition or other metrics are due to growing taxa.

L220 "and, at local scales, resource limitation." I think this clause is clearer with commas.

L223 "But even for drought-resistant taxa, growth was often compromised (Fig. 3B)." I'm not sure how I can discern this conclusion from Figure 3B. There is no indication of drought resistant or not resistant taxa in the figure (I don't think we can say that just because something grows in the drought treatment that it is resistant to drought). Further, I don't recall any definition of "drought resistant" taxa. I suppose one could take the classification of top 18-O assimilators in Fig 5, or the drought exclusive taxa in Fig 3A, but these were not presented as an indication of drought resistance. I also note that the term "resistant" only appears in the abstract and discussion. I suggest that if you would like to discuss the data this way that further analyses or at least a deeper exploration of the presented data are needed. E.g., present the argument that the top 18-O assimilators under drought are therefore "resistant", explain how "top assimilators" are selected (e.g., 95th percentile of growth?), and then present an analysis of the growth rates of those specific taxa under the different treatments, relative to the average growth response (or the growth response of "nonresistant" taxa).

L249 I find this to be a bit of a stretch. What was the average response of predatory bacteria relative to the overall community response, or relative to the growth weighted community size? Also, there were no statistics presented on predator response.

L252 I'm not sure what is meant by "partly higher relative growth rate" under future climate conditions in Fig 3B here when comparing drought response in ambient vs future climate (i.e., drought vs drought + FC presumably). The figure shows that the future climate + drought treatment had lower mean relative growth rate than drought alone.

L258 Relevant to the attenuation of drought response due to higher temperatures, do the heated plots experience lower moisture due to the heating that would precondition the community to more severe drought? It would be worthwhile to discuss the results in this context if so.

L266 I guess here you are addressing my comment directly above, but this is speculative and I am somewhat surprised that there are no longer term measurements of soil moisture from the future climate

treatments (or from the overall experiment in general, e.g., in a previously published paper). In fact, there are soil moisture measurements presented at line 342-343 in the context of the 18-O vapor equilibration. I would suggest at the very least further analyzing those soil moisture data to examine differences in moisture caused by the heating treatment alone compared to the control. If possible it would also be good to discuss any previously published changes in moisture content due to the heating treatment to put the hypothesis presented at L258 in context.

L266 "This might have indirectly selected for more drought-resistant taxa." To me a reduction in soil moisture seems like *direct* selection pressure in the context of drought resistance.

L286 What does "modified respiration" mean? Lower, higher?

L294-296 I'm not sure I agree with this wording. It's not that active taxa were "lost" under drought, but that only a subset of the total taxa were able to retain growth. Fig 1A clearly shows that the total community present was largely unaffected by drought, suggesting that there is potential for that community to recover if the drought is alleviated.

L295 What is meant by "low-diversity" here?

Reviewer #2 (Remarks to the Author):

*The article describes utilization of a long-term experiment manipulating drought stress and future climate conditions (elevated CO₂ ppm temperature) to examine how these two factors interact to alter microbial (bacterial and archaeal) growth and community composition. This is performed using a novel stable isotope labeling technique that leverages water amendment in the form of vapor so as to avoid known effects of adding liquid water to water-stressed soils. Results indicate that drought stress in particular constrict microbial community composition, and yet future climate conditions seem to reverse or mollify some of these effects.

*Article gives clearer insight into a topic that has much interest among likely readers. However, the study suffers from interpretability of some response variables (e.g., relative growth rates) in its current state (see comments below). Results and Discussion deserve to be more broadly covered in a longer-format publication, I believe, as there are many nuances which are hard to describe in an article of this length. Overall, the results confirm what prior understandings would have predicted concerning reductions in growth rate and constraining of diversity under moisture stress. While results will be of

interest to many readers, they are mostly 'novel' in respect to the use of the vap-qSIP method rather than transforming perspectives on microbial ecology.

*I also have reservations about the experimental design. It is unclear from the description whether all treatments (ambient, ambient + drought, future, future + drought) were subjected to the same 6-week rainout condition? Additionally, it's unclear whether 'drought' in the naming of treatments is referring to this short-term rainout treatment or any longer 6-year (since 2014) manipulation. On Lines 319-320 specifically, the authors indicate that the future climate plots had been maintained for 6 years. Have the drought plots not been maintained for this length of time? Overall it's unclear which of the treatment plots have a legacy (6 years) of treatment for drought. If only the future climate plots have a legacy of treatment, whereas the drought treatments do not, then this dramatically influences the interpretation of results as there would be pre-adaptation of communities to one stressor (future climate) but not the other (drought stress).

I believe the authors could address some of the above concerns (e.g., interpretability of RGR, experimental design and legacy treatments), yet others such as providing further space for description of Results and Discussion could conceivably not be accommodated. For that reason I believe the paper should be rejected. Additional line edits to improve the article are below.

Line 39: 'smaller' is used to indicate reduced biomass, or reduced abundance?

Line 39: 'active' according to what metric? Microbes having assimilated isotopes?

Lines 41-42: not immediately clear what is meant by 'modified the drought response, alleviating the loss of growing taxa within distinct communities.' Is this referring to some functional change that was noted in the simulated climate change? If so, functional characteristics should be noted for the 'control' drought conditions. Is 'growing taxa' a reference to greater activity of microbes under simulated climate change? It's also not clear from the statement what (if any) shifts in the microbial communities occurred under simulated climate change.

Line 43: unsure what is meant by 'pre-conditioning' in this context.

Line 55: seldom

Line 57: 'perform' is perhaps better replaced with 'function'

Line 58: unclear what 'predictions' are being referred to.

Line 72: this could be identified as the Birch effect.

Line 81: It seems there are four total treatment combinations, more than the three alluded to here (drought, future, drought + future).

Line 84: How does introduction of water vapor differ than introduction of liquid water, from a microorganism's perspective? Wouldn't the water vapor humidity condense on soil surfaces, effectively making liquid water available to a microbe in the same way that directly adding liquid water would? Perhaps the authors could mention what differences in growth rates and microbial activity look like when directly comparing water vapor and liquid water amendments to dry soils.

Line 88: Again, I think this description misses one of the treatment combinations (ambient)

Line 89: +3C relative to what?

Line 92: And functions, like growth rate!

Line 93: Unclear where the 'pre-adaptation' element is included in the experimental design so far described.

Line 105: "Total community composition shifted in response to drought". Is this relative to the 'ambient' treatment? If so, it would be helpful to indicate 'ambient' conditions were used as a control for comparison of effects.

Lines 107-108: Unclear what 'treatment conditions' are. Is this all of the treatments other than 'ambient'?

Line 108: Unclear what 'diverging abundances' means in this context. For Supp. Fig 1, do the author's mean diverging community composition?

Line 109: 'drought caused shifts in the growing and total community'

Line 115: It would be helpful to have a definition for ASV provided here, or earlier. Also, do the authors have justification for using ASV rather than OTU-level analysis? More information is not necessarily better information.

Line 116: What are the units for 0.05? percent?

Line 123: I think what is meant is the percent of total community that was growing? This is not clear from "the size of the growing community"

Line 129: The percent data would be better communicated as an effect size relative to ambient conditions, rather than mentioning the raw percentages for individual treatments (which don't mean much without the context of a control).

Line 146: Unclear what the difference between the two ASV numbers is. Were there only 5,116 total ASV's identified among all soils?

Lines 150: Again, I think it would be more informative to discuss effect sizes; by how much did drought decrease relative growth rates?

Line 164: Unclear why normality and homoscedasticity assumptions need to be met for a simple comparison of mean # of growing phyla in two treatments.

Line 198: "While ASVs classified as these putative taxa..."

Line 210: "...why results are often ambiguous..."

Lines 241-242: Not clear from the discussion what evidence points to either horizontal and/or vertical gene transfer, unless the authors mean to say that no support for either mechanism was directly supported.

Lines 311-320: Were all treatments (ambient, ambient + drought, future, future + drought) all subjected to the same 6-week rainout condition? Additionally, it's unclear whether 'drought' in the naming of treatments is referring to this short-term rainout treatment or the longer 6-year (since 2014) manipulation.

Line 344: Unclear which treatments 'respectively' received which ^{18}O enrichment levels.

Line 357: Supplemental Fig. 8 (not 5). How does the speed with which convergence on an average ^{18}O enrichment of soil water affect results? For the future climate + drought treatment, an average was reached within 50 hours but for the future climate treatment, ^{18}O enrichment of soil water was continuing to equilibrate (according to the model showing in Supp Fig. 8) up until through 100 hours of incubation.

Line 370: During fractionation?

Line 375: Fractions (not factions)

Fig. 2: (Panel C) Unclear how the relative growth rates are calculated such that all values are less than 1. From Line 342, is it to be assumed that these are percentages of maximum possible growth, where maximum possible growth would be an APE of taxon matches that of soil water APE?

Reviewer #3 (Remarks to the Author):

General comments: This is important research and the method used is novel. Understanding the response of soil microbes to multiple interacting global change factors is critical for predicting whether future ecosystems will be a source or sink for atmospheric carbon and is also critical for managing ecosystems to maintain soil fertility (i.e., maintaining healthy levels of soil organic matter). The method itself is important because it allows for measuring microbial growth in dry soils (i.e., under drought conditions). While the results appear sound and are exciting, the authors miss an important opportunity early in the paper to set up a strong rationale for their work. The writing is often vague and skips around such that there isn't a clear, linear storyline. The authors also overlook (or fail to appropriately highlight) important previous work on microbial responses to soil drying/drought. This paper is certainly not the first to look at this topic. What is new and exciting is that the authors were able to determine "Who" is active and under what conditions. The paper has potential to make a significant contribution to the

literature but needs to be reframed to meet that potential. I provide some specific comments below to assist the authors in revising the paper.

Specific comments:

L41-42. Ending of sentence (“...alleviating the loss of growing taxa within distinct communities.”) is unclear/vague. I realize that space is limited in the abstract but providing a more specific result would make the abstract clearer and more compelling. The last two sentences of the abstract are also quite general, and these could be combined into a single sentence to give more space for a specific result in the sentence above, while providing for a more concise, harder-hitting ending to the abstract.

L44. Why the specific focus on agriculture here. Certainly having drought resistant taxa in any ecosystem will become increasingly important for maintaining critical ecosystem functions.

L45. Awkward sentence construction: “...predicting future drought effects needs drought experiments...”.

Introduction: there is little information that puts soil C and its stabilization/loss into context. While I understand that context as a soil microbial ecologist, I think you need to provide that for the general readership of Nature Communications. That is, why care about soil C? How might it be impacted by global change? The same goes for drought. Can you provide some more specific information about the potential frequency of drought under climate change? All that to say, the introductory paragraphs are a little simplistic and vague. You’re missing an opportunity to build a strong rationale for your work.

L60-62. This is an interesting statistic, but it seems a given that most respiration would come from active microbes, and for a general audience who likely doesn’t know how many microbes (active or dormant) are in soils, this info isn’t that informative without more context.

L80-81. Is the method only possible for bacteria and archaea? Why weren’t fungi also evaluated? Are there limitations to the method in this regard?

L93. Define “pre-adaptation”. Are you using this in an evolutionary context? If not, I suggest using a different term.

L102. I don’t think “Non-metric Multidimensional Scaling” should be capitalized.

L105-108. Very neat and important result!

L206. Consider rewording this sentence. There's been extensive work on microbial responses to drought/drying (e.g., significant work on this topic area by Mary Firestone and Josh Schimel, among others), but what your study provides is a look at who's active and under what conditions. I think you need to do a better job both here and in the introduction of acknowledging previous work, while explaining the novelty of your own.

L305. What are the dominant plant species in the system? Do they vary across the experimental treatments? What is the soil type and general soil characteristics (i.e., texture, pH, C&N content)? This manuscript should include some basic information and not require the reader to go to another paper to find.

L324. Why 95°C when 105°C is standard for mineral soils?

L334-335. This seems like a very small sample size. Are larger samples not possible with this method? Did you evaluate the effect of sample size on outcome?

L360. By "snap-frozen", I assume you mean "flash-frozen"?

L375. "Fractions" not "factions".

REVIEWER COMMENTS

Reviewer #1 (Remarks to the Author):

NCOMMS-23-08901

The manuscript by Metze et al. examined the growth response of soil bacteria to a combination of six-week drought (i.e., drought manipulation) and 6-year elevated temperature and CO₂ manipulations (i.e., "future climate" manipulation). The authors found that previous exposure to future climate conditions attenuated the drought response. That is, while drought reduced the number of growing taxa by about 2/3 relative to ambient conditions, the number of growing taxa was only reduced by about 1/3 when the community was previously exposed to future climate with subsequent drought manipulation (Fig 2A). Further, the authors focused on growing taxa using 18-O qSIP modified with a previously described vapor equilibration technique that allows measurement of the growth response of microbes in dry soils without adding excess water (and thus affecting the growth response). While I am not an expert on soil water physics, and thus can not comment on the specifics of the vapor equilibration method, the technique certainly could provide an advance in understanding of microbial growth in dry soils. Given this, the authors show that while no changes in community composition due to drought were detected in their experiment using DNA alone, they did detect changes in community composition using the 18-O labeling vapor equilibration technique, pointing to a subset of taxa that were able to maintain growth even under a severe drought (Fig 1).

We thank the reviewer for their helpful comments which substantially increased the quality of the manuscript, particularly, regarding clarity, transparency, and consistency. We were able to address all comments and added extra figures, definitions, and statistics. We hope that we could clear all concerns and resolve unclear sections.

In general I find the paper to be well-written, however there are some points where I feel either clarification is needed or the presented data do not support the claims. For example, the authors discuss "drought resistant" taxa prominently in the abstract (L44) and in the

discussion (L222 and L267), however there was no formal analysis or definition of what the authors consider "drought resistant" taxa. While this is partly a matter of semantics, it is impossible for the reader to evaluate statements such as "even for drought-resistant taxa, growth was often compromised" (L222) without knowing which taxa the statement applies to. There are data presented that could potentially be used to classify drought resistance, such as the taxa that maintain high growth under drought (Fig 5B) or the taxa that were unique to the drought treatment (Fig 3A Venn diagram), but I feel a further analysis and clear explanation of what the authors consider to qualify as drought resistance is needed to support this. E.g., how were the "top 18-O assimilators" in Fig 5B selected?

We agree that a more careful definition of a microorganism's ability to endure drought was necessary. Drought resistance can be divided into drought avoidance and drought tolerance strategies. While drought tolerance entails physiological mechanisms that allow a microorganism to be active and even grow in dry soils, drought avoidance strategies mostly consist of entering dormancy. Here, we focused on the bacteria and archaea that managed to grow in drought-affected soils and defined those as drought-tolerant (L: 135, 318). The term "resistant" was not used anymore since it also includes taxa persisting in a dormant stage which cannot be differentiated from inactive or starving using qSIP. We also added a more detailed description of the selection of the top 18O assimilating taxa in the methods section, results, and figure captions. See L: 227, 534-540, 871.

In addition there are some claims that do not appear to be supported by the data as presented, specifically, 1) at L148-150 I can not sum the subsections of the Venn diagram (Fig 3A) to find the figures the authors describe and so I question whether the numbers should be rechecked or the wording clarified, 2) the sections on bacterial predators at L197-201 (results) and L249 (discussion) does not appear to be supported by any statistics or in-depth analyses, and so it is hard to find much support for the conclusions drawn. Finally, the authors discuss the attenuation of drought response in the context of future climate as a possible response to reduced moisture levels in the heated treatments (L263-267). I agree that this is a distinct possibility. However, the authors do not prominently present any data on soil moisture, except for as an aside at a different point in the discussion (L278-279), or discuss any previously published soil moisture data from the experiment. I suggest featuring

a discussion about any available data on soil moisture as this will lend credence to the conclusions drawn from the current data.

We thank the reviewer for addressing these points since they were important to increase the credence of the presented manuscript. We revisited the venn diagram (Fig. 3A) and confirm that all given numbers were correct and supported by our data. However, we agree that the description of the treatment comparisons was not sufficiently clear. Hence, we rewrote the section to provide more detail and avoid possible misinterpretations (L: 177-190). Regarding the section about predators, we added an extra figure (Supplementary Fig. 9) and statistics to support our conclusions as well as extended the respective paragraph in the results part including a more comprehensive lead-in (L: 243-259).

We thank the reviewer, in particular, for stressing that we missed an opportunity to make a stronger argument about the attenuation of the drought response due to lower soil moisture contents in the simulated future climate plots. There are long-term measurements available from previous studies, supporting this claim. We cited these studies and rewrote the respective sentences (L: 331-335). We also analyzed the soil moisture data gathered in the frame of this study, confirming previously found trends (L: 117).

I would also suggest at least one modification to one of the primary statistical methods: the authors computed community dissimilarity based on Euclidean distance (Fig 1). However, this distance metric is widely considered to be inappropriate for ecological community composition analyses (e.g., McCune and Grace, Analysis of Ecological Communities, 2002). Unless the authors can justify this choice I would suggest re-analyzing the data using an appropriate distance metric.

The reviewer raised an important point, and we totally agree with them when it comes to working with microbial community data based on relative abundances. However, according to a new, recently published, quantitative sequencing framework (<https://www.nature.com/articles/s41467-020-16224-6>), it is more appropriate to compute Euclidean distances after centered log-ratio (clr) or log-ratio transformation when dealing with absolute abundance measurements inferred by amplicon sequencing and digital droplet PCR. Since the presented data are based on absolute abundance measurements, we argue that it is more appropriate to use Euclidean distances. Nonetheless, we agree that our

approach was inconsistent, and we did not use the appropriate visualization method. Hence, we re-computed all distances after clr transformation and visualized them using principal component analysis (PCA). We agree that another distance metric might have been more suitable for the PCA performed on relative growth rates instead of absolute abundances (Supplementary Fig. 1) but since we aimed to directly compare the results presented in Fig. 1 and Supplementary Fig. 1, we considered it to be better to use the same analyses pipeline.

More details on these comments and additional comments are given below.

Detailed comments:

L412 I suggest relating this back to the 15-16 density fractions per sample cited above at L375. I.e., how many fractions result from filtering to the 1.614-1.753 density range? You might also consider giving some justification for why this filtering was performed for folks who are not familiar with qSIP protocols.

Thank you for this suggestion. The density filter was applied to exclude low-density fractions that were contaminated with the fractionation medium, which was water in our case, resulting in the loss of 0-3 fractions per sample. No fractions were lost due to the upper threshold which marks the upper density of the CsCl gradient in our experiment. We clarified this in the manuscript. See L:496-498.

L426 It seems the sample/fraction number cutoff must be an integer. I suggest giving the minimum cutoff instead of "at least 25% of all fractions (~4 fractions)".

The fraction number cutoff was 4 fractions. We clarified this in the manuscript in L: 510.

L429 I suggest providing here the range of APE 18-O and the number of taxa detected with less or greater than the cutoff given the prior filtering steps.

Prior to applying the minimum enrichment threshold (APE 18O > 0.05), the number of growing taxa (growth defined as APE 18O > 0) was 5,652 ASVs with a minimum APE 18O of zero. We included this information as well as the max. APE 18O in the manuscript, which remained unaffected by the cutoff. 536 ASVs were lost after applying the enrichment cutoff (Before: 5652 ASVs; After: 5116 ASVs) See L: 512.

L457 This is unclear. What was the response matrix for calculating Euclidean distance? Growth rates? Presence-absence in either the growing or nongrowing matrix? Further, Euclidean distance is typically considered unsuitable for community data due to a high proportion of absences (and the assumption that zeros can be either real or present but undetected with no way to differentiate these two scenarios). Depending on what the input data was I suggest either adding some text to justify the use of Euc distance or recalculating with Bray-Curtis distance or another distance metric that is suitable for community data. This affects both PERMANOVA statistics and the NMDS.

We computed Euclidean distances using the absolute abundances of all taxa (Fig. 1A), the absolute abundances of all growing taxa (Supplementary Fig. 1, revised: Fig. 1B), and the relative growth rates (RGR) of all growing taxa (Fig. 1B, revised: Supplementary Fig. 1). We aimed to show that future climate conditions as well as their interaction with drought have a significant effect on the community composition of the active community, an effect masked at the total community level. By also using growth rates, we aimed to show that this was not only reflected by the abundance of active taxa but also by their growth. We acknowledge that this was not sufficiently well explained and might not have become clear by the way we have chosen the figures for the main body of the manuscript. Thus, we exchanged Supplementary Fig. 1 and Fig. 1B so that all panels of Fig. 1 are based on absolute abundances, visualizing shifts in community composition. We agree with the reviewer that Euclidean distances are often considered unsuitable when working with microbial community data consisting of relative abundances. Here, however, we dealt with absolute abundances and, as explained in the answers to the previous comment,, followed the guidelines of a new quantitative sequencing framework using log/center log ratio (clr) transformations followed by the computation of Euclidean distances, visualized by either NMDS or PCA. For reasons of comparison, we have chosen the same rationale for the PCA using growth rates instead of absolute abundances. To increase the consistency, we now also used the same transformation method (clr) for all data types (absolute abundances, RGRs). We tried to make this clear in L: 549, 554,557.

L469 What is meant by "top 18-O assimilating taxa"? Is this referring to the filtering previously described? Please clarify how these taxa were selected and/or what was the cutoff.

The top 18-O assimilating taxa were identified by ranking growing ASVs based on their proportional 18-O assimilation. We estimated proportional ¹⁸O-assimilation for each growing ASV by re-calculating their relative abundances (sum of relative abundances of growing taxa = 1) and multiplying them by their relative growth rate (RGR). These weighted enrichment values were then divided by their sum, producing a proportional ¹⁸O-assimilation value ranging between 0-1. This value was calculated because although certain taxa might be growing faster than others, their contribution to the overall community-level growth will also depend on their abundance. The top five 18-O assimilating ASV, as depicted in Fig. 5, were simply the ASVs with the highest proportional 18-O assimilation per sample. Since an ASV can be amongst the top assimilators in different samples, the overall number of the top five assimilating ASVs varied between samples exposed to ambient precipitation (24 ASVs) and drought (18 ASVs). We choose to visualize the pool of the top five taxa to limit the heatmap to an easily understandable size. We added more descriptions in the results and methods section to clarify this point. See L: 224-231, 534-540.

L100-110 As noted above, Euclidean distance does not appear to be appropriate for this data set. While it was not entirely clear in the methods, based on the description of the results it seems this is community composition. I suggest recalculating with an appropriate distance metric. Further, there is really no need to use NMDS with Euclidean distance, just use PCA which is easier to interpret (axis eigenvalues).

We used PCA instead of NMDS in the revised version of the manuscript to visualize Euclidean distances which were chosen, as described above, due to using absolute abundance data.

L106-110 The effects on active community composition could be discussed with more nuance. The PERMANOVA statistics in Supp. Fig 1 clearly show that drought had by far the greatest impact on activity community 16S abundance (3x higher R²), but also that the

other effects were marginal ($P > 0.05$). In Fig 1B twice as much variation is explained by drought as climate or their interaction.

We thank the reviewer for this comment. As described above, we consistently clr transformed all community data (absolute abundances and relative growth rates) and performed PCAs. For direct comparison, we shifted the PCA showing the active community composition based on absolute abundances into the main manuscript and moved the PCA showing the active community composition based on relative growth rates in the supplementary material. Due to these changes we also re-run PERMANOVAs which showed that treatment effects on the active community were similar comparing PCAs based on either growth or abundance data. We agree that the nuances of the PERMANOVA results were not sufficiently discussed and elaborated in L: 123-131.

L117 Picky, but I suggest separating the statistics for the different model terms with a semicolon instead of a comma. It is easy to get lost in the long string of model terms. E.g., Drought: model coefficients, etc; Climate: model coefficients). This goes for all of such reported statistics.

We incorporated the suggested changes here and elsewhere.

Figure 2. Because you are performing mean comparisons I think the error bars in panels B and C should be changed to standard errors (an estimate of the accuracy of the sample mean relative to the population mean) as opposed to std deviation (variability of the sample).

We changed the error bars in Fig. 2B and Fig. 2C to standard errors.

L148-150 I think this section should either be clarified or the numbers rechecked. I can not come up with the same numbers as reported based on my understanding of Fig 3A.

- E.g., "drought only" (L148) is reported as 184 ASVs, however I interpret this as ASVs only found in drought treatment which is shown as 233 in the figure.

- "Twice as many taxa were shared between the future climate and drought treatment (374

ASVs)" I interpret as the overlap between

"Ambient + drought" and "Future climate" treatments, however these sum to 113 ASVs.

- Alternatively the whole sentence "twice as many taxa were shared between the future climate and drought treatment (374 ASVs) and the ones exposed to future climate and ambient precipitation." could also be interpreted as the overlap between "future climate + drought" and "future climate" treatments in the figure although I would then expect the number of ASVs shared to be reported at the end of the sentence. In any case the overlap between the FC and FC + drought treatment sums to 241 ASVs instead of 374.

- Further, partly because the comparisons being made are unclear, it is also unclear to me why these particular comparisons are of interest. It would help to provide a sentence outlining the rationale behind the comparisons being made, e.g., " we specifically compared overlap in growing taxa in the drought + ambient climate treatment to the future climate treatment and the FC + drought treatment to discern which taxa were responding to different pressures...."

We thank the reviewer for pointing out that this section was unclear and that the numbers did not seem to add up based on the text. The rationale behind this figure was to show two things: First, the majority of ASVs were specifically growing in only one treatment each, potentially indicating their preferred growing conditions or the conditions at which they are most competitive. Second, we wanted to visualize that several taxa active in treatments not exposed to drought (ambient, future climate) are still growing in drought-affected soils but this was true for more taxa if soils have been exposed to future climate conditions as well. Thus, the number of shared taxa in the manuscript is calculated by adding the number of taxa shared between each drought treatment and the other two treatments not exposed to drought:

- *Overlap: Future climate & drought vs. ambient and future climate = 374 ASVs = $79+17+32+113+115+18$)*
- *Overlap: Drought vs. ambient and future climate = 184 ASVs = $32+17+18+32+32+53$)*

We agree with the reviewer that this was not sufficiently clear and rewrote the section including a description of why this comparison was of interest. Changes made can be found in L: 177-190.

L153 Why does it seem like some statistics are missing here? The parentheses end with calling out a panel but no stats are given.

This is correct. We thank the reviewer for noticing this mistake. The statistics were missing because the previous sentence referred to drought effects compared to ambient precipitation, whereas the right center panel of Figure 3 aims to visualize the impact of future climate conditions on the growth of taxa shared between both drought treatments (Ambient + Drought, Future Climate + Drought). We clarified this and added an extra sentence in L: 199.

Fig 4. I suggest modifying the x-axis label to read "Relative change in ASV richness"

We thank the reviewer for this suggestion. We changed the x-axis label accordingly.

Fig 5. You might consider adding an indication of the phylum to the figure as phylum is discussed prominently in the text. Also, how were "top 18-O assimilators" defined for this figure? E.g., one panel has fifteen rows by my count and the other has twelve. Given that "ASV identities were agglomerated" at the genus, family, or phylum level, are we to understand that these designation have not only variable numbers of ASVs in them, but also could represent very different levels of taxonomic cohesion? I think a better description of data treatment for this figure would be helpful.

Based on the reviewer's comment we added phylum names to Fig. 5, Supplementary Fig. 5, and Supplementary Fig. 6. In this figure, we aimed to visualize the top 18O assimilating taxa, representing dominant contributors to the total community's growth based on their growth and abundance. After calculating proportional 18O assimilation for each ASV in each sample, we ranked them and selected the top five ASVs. Due to the overlap in top 18O assimilating taxa between samples, this resulted in 26 ASVs in drought-unaffected soils (ambient, future climate) and 19 ASVs in drought-affected soils (drought, drought & future climate). ASVs were agglomerated at the genus level, however, if no taxonomic assignment was available, there were agglomerated at the family or phylum level. We clarified the selection of top taxa

and the agglomeration at the genus/family/phylum level in the figure captions (L: 871), results section (L: 226), and methods (L: 534).

Further, I would suggest presenting all of the data behind this figure, either at the agglomerated taxonomic level, the ASV level, or both, in a heatmap(s) in the supplement (i.e., include all taxa). You could highlight the subsections of data you are pulling out for the main text. That way the reader can make comparisons across the entire dataset if desired instead of focusing only on the particular taxa the authors choose to highlight.

We thank the reviewer for this comment and included two new figures in the Supplementary Material (Supplementary Fig. 5, Supplementary Fig. 6), presenting a more extensive dataset behind Fig.5 of the manuscript. Unfortunately, presenting all data behind this figure goes beyond comprehensive visualization in the form of heatmaps since that would include ~1300 - 2700 ASVs. Thus, we decided to visualize the pool of the top 50 ASVs per sample in drought-affected and drought-unaffected soils, representing 200 ASVs and 217 ASVs, respectively. These ASVs accounted for 49.7 ± 16.4 % and 35.4 ± 6.4 % of the community growth under ambient precipitation and drought. We referred to these figures in L: 227.

L197-201 This bit about bacterial predators does not have much (any?) lead-in. Could you provide a small bit of background and/or some methodological info to describe how this was done? Further, I would like to see some statistics here instead of just a heatmap as there is a paragraph in the discussion section on these responses.

We agree with the reviewer that context, as well as statistical support, were missing for this section. Thus, we performed statistics, rewrote the section, elaborated more on the rationale and methodology, and added a new figure to the Supplementary Material (Supplementary Fig. 9A & B), showing the decrease of proportional 18O assimilation and abundance of putative predatory taxa with drought including statistics. See L: 243-260.

L211 I think a better job could be done with appropriate citations here. E.g., the Malik paper performed mRNA sequencing and so specifically targeted expressed transcripts and not DNA (and therefore does not support the preceding statement at L210). The Evans paper

examined the effects of temporal rainfall intensity (i.e., fewer, larger rainfall events vs more frequent, smaller events), and not drought, per se, and further primarily focused on an in vitro wet up experiment over the course of a 115 day time series — where we can reasonably assume that changes in community composition or other metrics are due to growing taxa.

We agree that the citations did not seem to fit the context, especially regarding the Malik paper. We wanted to stress that drought effects on diversity are often ambiguous which is supported by the Malik paper. However, we understand that the previous sentence suggests that we are referring to DNA-based papers. Hence, we removed the Malik as well as the Evans paper from the citations and selected additional appropriate references (https://bsssjournals.onlinelibrary.wiley.com/doi/full/10.1111/ejss.12429?saml_referrer, <https://www.nature.com/articles/srep34434>, <https://www.sciencedirect.com/science/article/abs/pii/S0929139314001772>). See L: 271-272.

L220 "and, at local scales, resource limitation." I think this clause is clearer with commas.

We adopted this suggestion.

L223 "But even for drought-resistant taxa, growth was often compromised (Fig. 3B)." I'm not sure how I can discern this conclusion from Figure 3B. There is no indication of drought resistant or not resistant taxa in the figure (I don't think we can say that just because something grows in the drought treatment that it is resistant to drought). Further, I don't recall any definition of "drought resistant" taxa. I suppose one could take the classification of top 18-O assimilators in Fig 5, or the drought exclusive taxa in Fig 3A, but these were not presented as an indication of drought resistance. I also note that the term "resistant" only appears in the abstract and discussion. I suggest that if you would like to discuss the data this way that further analyses or at least a deeper exploration of the presented data are needed. E.g., present the argument that the top 18-O assimilators under drought are therefore "resistant", explain how "top assimilators" are selected (e.g., 95th percentile of growth?), and then present an analysis of the growth rates of those specific taxa under the

different treatments, relative to the average growth response (or the growth response of "nonresistant" taxa).

We thank the reviewer for drawing our attention to the usage of the term "drought-resistant". As described above, drought resistance is divided into drought tolerance and drought avoidance strategies with the latter being primarily dormancy. Drought tolerance, on the other hand, entails physiological adaptations that allow microorganisms to remain active and growing during drought. Since qSIP does not indicate which taxa are dormant but only which taxa are growing (growth is defined as replication), we replaced "drought-resistant" with "drought-tolerant" in the revised version of the manuscript. We defined drought-tolerant taxa as those who possess the physiological adaptations to allocate energy to growth even under drought conditions and, hence, still show growth (L: 135, 318). In addition, we stressed when taxa grew at ambient as well as drought conditions by using the terms "drought-persisting" or "drought-enduring (Fig. 3 B). "Drought-tolerant" or variations of this term have been used before in the context of soil microbial ecology (doi:10.1038/nrmicro.2017.16., <https://doi.org/10.1146/annurev-ecolsys-110617-062614>). Based on this, we revised the rest of the manuscript. We defined drought-tolerant and used it where appropriate.

L249 I find this to be a bit of a stretch. What was the average response of predatory bacteria relative to the overall community response, or relative to the growth weighted community size? Also, there were no statistics presented on predator response.

Based on the previous reviewer's comment, we added additional figures and statistics about putative bacterial predators (Supplementary Fig. 9A & B, L: 243-260). Based on this data, ~90% of putative bacterial predators stopped growing during drought (total community ~50%) associated with a > 90% reduction of their abundance. When including non-growing putative predators, drought also decreased their mean relative growth rates as presented in the figure below (Statistics: Two-way ANOVA). Therefore, we argue that drought causes a strong reduction in predation via predatory bacteria which, regarding the loss of active taxa, was even stronger as compared to the average of the total community. Though, we

acknowledge that we can only speculate about the reasons behind this observation. We tried to stress this in L: 313-317.

Figure 1: Mean relative growth rates of putative predatory bacteria across treatment conditions

L252 I'm not sure what is meant by "partly higher relative growth rate" under future climate conditions in Fig 3B here when comparing drought response in ambient vs future climate (i.e., drought vs drought + FC presumably). The figure shows that the future climate + drought treatment had lower mean relative growth rate than drought alone.

We apologize for the misunderstanding. By "partly higher relative growth rate", we referred to the effect at ambient precipitation where future climate conditions led to an increase in relative growth rates of ASVs active in both treatments (ambient, future climate). We clarified this in the text and referred to the respective figure panel. L: 320-321.

L258 Relevant to the attenuation of drought response due to higher temperatures, do the heated plots experience lower moisture due to the heating that would precondition the community to more severe drought? It would be worthwhile to discuss the results in this context if so.

We discussed this point in more detail below.

L266 I guess here you are addressing my comment directly above, but this is speculative and I am somewhat surprised that there are no longer term measurements of soil moisture from the future climate treatments (or from the overall experiment in general, e.g., in a previously published paper). In fact, there are soil moisture measurements presented at line 342-343 in the context of the 18-O vapor equilibration. I would suggest at the very least further analyzing those soil moisture data to examine differences in moisture caused by the heating treatment alone compared to the control. If possible it would also be good to discuss any previously published changes in moisture content due to the heating treatment to put the hypothesis presented at L258 in context.

We thank the reviewer for pointing out that we missed an opportunity to make a stronger point here. There are long-term measurements of soil moisture available (taken over several months) that show how future climate conditions substantially decreased soil moisture even before drought simulations (<https://www.nature.com/articles/s41396-020-00735-7#Sec2>, <https://www.sciencedirect.com/science/article/pii/S0038071721003102>). We also analyzed the soil moisture data from the end of our drought experiment and found a small but significant effect of future climate conditions on soil moisture. We included this information in the results section and reframed the respective section in the discussion. Please find the made changes in L: 331-335.

L266 "This might have indirectly selected for more drought-resistant taxa." To me a reduction in soil moisture seems like *direct* selection pressure in the context of drought resistance.

We agree with this comment and removed "indirect" L: 334.

L286 What does "modified respiration" mean? Lower, higher?

We thank the reviewer for pointing out this case of vague language. We changed "modified respiration" to "slightly higher respiration". L: 354.

L294-296 I'm not sure I agree with this wording. It's not that active taxa were "lost" under drought, but that only a subset of the total taxa were able to retain growth. Fig 1A clearly shows that the total community present was largely unaffected by drought, suggesting that there is potential for that community to recover if the drought is alleviated.

We agree that the use of the word "loss" was misleading. We changed the wording and stressed that many taxa active at ambient conditions stopped growing under drought (L: 366.). Though, we would like to stress that the total community composition (Fig 1A) was significantly affected by drought, limiting the potential for recovery if drought is alleviated.

L295 What is meant by "low-diversity" here?

As mentioned above, we wanted to underline that many taxa active at ambient conditions stopped growing under drought, strongly reducing the diversity of the active community in drought-affected soils. We agree that the term "low-diversity" was not sufficient to make this clear and elaborated more on this in the text. L: 366-369.

Reviewer #2 (Remarks to the Author):

*The article describes utilization of a long-term experiment manipulating drought stress and future climate conditions (elevated CO₂ ppm temperature) to examine how these two factors interact to alter microbial (bacterial and archaeal) growth and community composition. This is performed using a novel stable isotope labeling technique that leverages water amendment in the form of vapor so as to avoid known effects of adding liquid water to water-stressed soils. Results indicate that drought stress in particular constrict microbial community composition, and yet future climate conditions seem to reverse or mollify some of these effects.

We thank the reviewer for their comments and suggestions. They pointed out important sections where descriptions were unclear. We addressed these comments and changed several sections in the revised version of the manuscript.

*Article gives clearer insight into a topic that has much interest among likely readers. However, the study suffers from interpretability of some response variables (e.g., relative growth rates) in its current state (see comments below). Results and Discussion deserve to be more broadly covered in a longer-format publication, I believe, as there are many nuances which are hard to describe in an article of this length. Overall, the results confirm what prior understandings would have predicted concerning reductions in growth rate and constraining of diversity under moisture stress. While results will be of interest to many readers, they are mostly 'novel' in respect to the use of the vap-qSIP method rather than transforming perspectives on microbial ecology.

We addressed the reviewer's comments to improve the clarity and transparency of the calculations and interpretation of the response variables. Furthermore, we extended the results section and described the calculation of relative growth rates in more detail (L: 517). Thus, we believe that the revised version now includes all the important information to interpret the presented data and covers its nuances in more detail. Using vap-qSIP allowed us, for the first time, to investigate the actively growing community of bacteria and archaea

in drought-affected soils without changing the soil water content. It is these taxa and their activity that are considered to be the main contributors to biogeochemical processes in soil. We believe these results are novel and important since it has not been possible, thus far, to measure taxon-specific microbial growth in dry soils using standard methods since they always involve liquid water additions that constitute a rewetting event associated with an increase in growing microbes. While we agree that our data support previous findings, e.g. the drought-tolerance of many Actinobacteriota, previous findings based on amplicon sequencing data were often ambiguous because of the stability of extracellular DNA, also regarding diversity, potentially masking drought effects. We believe that our data resolves much of this uncertainty while providing new insights into the underlying growth responses of individual taxa and their drought tolerance. In addition, the effects of future climate conditions were not visible at the total community level, underlining the importance of using vap-qSIP to discover shifts in the ecologically most relevant fraction of the microbial community.

*I also have reservations about the experimental design. It is unclear from the description whether all treatments (ambient, ambient + drought, future, future + drought) were subjected to the same 6-week rainout condition? Additionally, it's unclear whether 'drought' in the naming of treatments is referring to this short-term rainout treatment or any longer 6-year (since 2014) manipulation. On Lines 319-320 specifically, the authors indicate that the future climate plots had been maintained for 6 years. Have the drought plots not been maintained for this length of time? Overall it's unclear which of the treatment plots have a legacy (6 years) of treatment for drought. If only the future climate plots have a legacy of treatment, whereas the drought treatments do not, then this dramatically influences the interpretation of results as there would be pre-adaptation of communities to one stressor (future climate) but not the other (drought stress).

We expanded the description of the experimental design in the revised manuscript version (L:105,106,110,397-401,). Only the drought treatments (ambient + drought, future climate + drought) have been exposed to the simulated summer drought event, lasting approximately 6 weeks. Future climate conditions have been manipulated since 2014. Plots of the drought treatments have experienced two summer drought simulations before 2020, namely in 2017

and 2019. All plots have, thus, been maintained for the same duration but drought plots, in addition to the climatic conditions, also experienced simulated summer droughts. The frequency of drought events in drought-exposed plots did not differ between climatic conditions. Future climate plots (including future climate + drought) had a 6-year legacy of future climate conditions which is why we argue that taxa in these treatments might have been pre-conditioned to lower soil moisture contents induced by higher temperatures and, hence, showed an alleviated drought response. We elaborated on this in the discussion and referred to soil moisture data from previous studies (L:331-335).

I believe the authors could address some of the above concerns (e.g., interpretability of RGR, experimental design and legacy treatments), yet others such as providing further space for description of Results and Discussion could conceivably not be accommodated. For that reason I believe the paper should be rejected. Additional line edits to improve the article are below.

We understand the reviewer's suggestion for a more nuanced description of the results and a more detailed discussion. Although a publication of this format is comparatively short, we believe that we could address these concerns and present an extended manuscript with more in-depth descriptions of methods, results, and discussion. We also elaborated on the calculation of relative growth rates, experimental design, and treatment legacy to resolve the remaining uncertainties.

Line 39: 'smaller' is used to indicate reduced biomass, or reduced abundance?

"Smaller" is used to indicate smaller growing communities regarding the percentage of the total community that is actively growing. This can be considered as an indicator of abundance. We re-wrote the abstract using less vague language and removed the term "smaller".

Line 39: 'active' according to what metric? Microbes having assimilated isotopes?

This is correct. Based on 18O-H2O quantitative stable isotope probing, taxa are considered growing and, hence, active if they incorporated the heavy isotope of oxygen (<https://journals.asm.org/doi/10.1128/AEM.02280-15>). We clarified that qSIP allows us to measure taxon-specific microbial growth in L: 36

Lines 41-42: not immediately clear what is meant by ‘modified the drought response, alleviating the loss of growing taxa within distinct communities.’ Is this referring to some functional change that was noted in the simulated climate change? If so, functional characteristics should be noted for the ‘control’ drought conditions. Is ‘growing taxa’ a reference to greater activity of microbes under simulated climate change? It’s also not clear from the statement what (if any) shifts in the microbial communities occurred under simulated climate change.

In our experiment, summer drought was simulated for 6-weeks in plots that were exposed to either an ambient climate or a simulated future climate (+3°C of warming above ambient temperatures and +300 ppm of CO₂ above ambient CO₂ concentrations) since 2014. 18O-H2O-qSIP is a method that allows for identifying actively growing microbial taxa and quantifying their growth rates based on the incorporation of 18O-H2O into their DNA during replication. Microorganisms use the 18O derived from water during replication and DNA synthesis. The higher the 18O enrichment of the DNA, the higher the inferred growth rate. Hence, all growing taxa are active and growth rates are a measure of their activity. However, not all active taxa are able to grow, for instance, if don’t possess the traits to allocate enough energy to growth under unfavorable conditions. These taxa just persist but could be translationally active. Nonetheless, their contribution to biogeochemical transformation processes is thought to be highly reduced (<https://doi.org/10.1007/s11104-022-05382-9>). Comparing the drought response of the growing community inferred by vap-qSIP under an ambient and future climate, we found more growing taxa under future climate conditions in growing communities distinct from the drought-only treatment in their composition. Hence, here we are referring to the number (richness) of growing taxa and the composition of the growing community. We re-wrote the abstract using less vague language and hope we could resolve unclear sections.

Line 43: unsure what is meant by 'pre-conditioning' in this context.

By 'pre-conditioning' we referred to the pre-exposure of future climate plots to higher temperatures and CO2 conditions for the past six years which exposed the soil microbiome to lower soil moisture conditions as well. By now mentioning the length of duration of the exposure of the plots to future climate conditions and using the term "pre-exposure", we tried to put this into context. See L: 41

Line 55: seldom

The corresponding section was re-written in the revised version of the manuscript without using the term "seldom".

Line 57: 'perform' is perhaps better replaced with 'function'

The suggested change was adopted. L: 86.

Line 58: unclear what 'predictions' are being referred to.

We clarified this with 'predictions of ecosystem processes such as carbon fluxes' in L: 87.

Line 72: this could be identified as the Birch effect.

The suggested change was adopted, and we referred to the Birch effect in L: 81.

Line 81: It seems there are four total treatment combinations, more than the three alluded to here (drought, future, drought + future).

The suggested change was adopted, and we added "ambient conditions" to the enumeration in L: 97.

Line 84: How does introduction of water vapor differ than introduction of liquid water, from a microorganism's perspective? Wouldn't the water vapor humidity condense on soil surfaces, effectively making liquid water available to a microbe in the same way that directly adding liquid water would? Perhaps the authors could mention what differences in growth rates and microbial activity look like when directly comparing water vapor and liquid water amendments to dry soils.

The water vapor equilibration method only seeks to isotopically enrich soil water. Condensation was never observed when using the water vapor equilibration method here and before (<https://onlinelibrary.wiley.com/doi/full/10.1111/qcb.15168>). The latter study also compared the effects of direct water application and the water vapor equilibration method for estimating microbial growth and respiration in dry soils. The study found that growth was overestimated by up to 250% and respiration by up to 500% when using direct water addition, artificially simulating a rewetting event. In contrast, the water vapor equilibration method does not require adding labeled water directly to the sample, leaving the soil water content unchanged which does not boost microbial activities induced by rewetting. We added this information to L: 94.

Line 88: Again, I think this description misses one of the treatment combinations (ambient) *The suggested change was adopted, and we referred to the respective control treatments in L: 106.*

Line 89: +3C relative to what?

We aimed to say: +3C above ambient temperatures and added this information in L: 105.

Line 92: And functions, like growth rate!

We included this comment in the following sentence. L: 111.

Line 93: Unclear where the 'pre-adaptation' element is included in the experimental design so far described.

A possible pre-adaption of the community was not part of the experimental design. 'Pre-adaptation' was changed to 'pre-exposure' which refers to the previous exposure (6 years) of soil microbes to future climate conditions before being exposed to drought as future climate conditions were characterized by 3°C warmer air temperatures which resulted in repeatedly drier conditions even before the drought simulation (L: 118-120, 332-336).

Line 105: "Total community composition shifted in response to drought". Is this relative to the 'ambient' treatment? If so, it would be helpful to indicate 'ambient' conditions were used as a control for comparison of effects.

We used two-way permutation-based multivariate analysis of variance (PERMANOVA) testing a full two-factorial design (Drought Yes, Drought No, Climate: Ambient, Climate: Future Climate) on the Euclidean distances computed using the absolute abundances to test for drought and future climate effects including their interaction. Drought effects on community composition were significant and can be found as inset panels in Fig. 1A.

Lines 107-108: Unclear what 'treatment conditions' are. Is this all of the treatments other than 'ambient'?

Here we referred to all four treatment conditions. We clarified this in L: 129.

Line 108: Unclear what 'diverging abundances' means in this context. For Supp. Fig 1, do the author's mean diverging community composition?

Community composition was compared using the absolute abundances of individual taxa. We agree that the wording 'diverging abundances' was unclear and used community composition instead.

Line 109: 'drought caused shifts in the growing and total community'

We thank the reviewer for pointing this out and clarified this in L: 132-134.

Line 115: It would be helpful to have a definition for ASV provided here, or earlier. Also, do the authors have justification for using ASV rather than OTU-level analysis? More information is not necessarily better information.

We agree with the reviewer that our definition of ASV (amplicon sequence variant) was not given in this section of the original version of the manuscript and have now included a definition of the term (see L: 139). We understand and interpret ASVs as unique sequences as denoised and processed with DADA2 using the standard settings. This means that each ASV retrieved from a soil sample represents a bacterial taxon. We understand that there are limitations to these assumptions (similar to OTUs, e.g. as discussed here: <https://www.fiererlab.org/blog/archive-lumping-versus-splitting-is-it-time-for-microbial-ecologists-to-abandon-otus>) and decided to report our 16S rRNA sequencing data as ASVs instead of clustering them into OTUs for the following reasons:

1) Bacterial genomes only harbor up to a few rRNA operons per cell which otherwise could inflate diversity estimates. Instead, the usage of ASVs has shown to produce similar or lower richness levels than OTU clustering (e.g. [10.7717/peerj.5364](https://doi.org/10.7717/peerj.5364) or <https://doi.org/10.1371/journal.pone.0264443>) which makes us confident that the diversity levels reported in our manuscript are robust.

2) The main goal of this manuscript was to identify bacteria that grow under the tested experimental conditions. ASVs provide an improved taxonomic resolution (as compared to OTUs) and allowed us to delineate distinct responses to drought under ambient and future climate between individual strains;

3) Using ASVs instead of OTUs makes microbial community data more comparable across studies. Each ASV is characterized by a unique sequence that can be identified in other studies, allowing for direct comparisons and cross-synthesis studies.

Line 116: What are the units for 0.05? percent?

Here, 0.05 refers to APE-18O which is an abbreviation for 18O atm percent excess. The excess quantity of a stable isotope, stated in atom percent excess, by which a stable isotope in a sample surpasses its presence in a reference, serves as a gauge for assessing its abundance. Thus, the unit for 0.05 is percent but converted to decimal. We acknowledge that this might be confusing and used the percent value (5 %) instead. This was adopted in L: 141.

Line 123: I think what is meant is the percent of total community that was growing? This is not clear from “the size of the growing community”

Yes, we referred to the percent of the total community that was growing and clarified this in L: 149.

Line 129: The percent data would be better communicated as an effect size relative to ambient conditions, rather than mentioning the raw percentages for individual treatments (which don't mean much without the context of a control).

We agree that effect sizes are a powerful tool to visualize shifts with regard to one control treatment. Here, we aimed to display differences between different treatments though, for instance, ambient vs. drought, drought vs. future climate & drought, or future climate vs. future climate & drought. Hence, we considered effect size not to be the optimal way of communication. We agree, however, that percentages are less meaningful without context data. Since total 16S rRNA gene copies did not significantly differ between treatments (Two-way ANOVA; Drought: $F = 0.7$, $df = 1$, $p = 0.39$; Climate: $F = 2.6$, $df = 1$, $p = 0.12$; Drought x Climate: $F = 0.004$, $df = 1$, $p = 0.95$), we consider the percentage values to be comparable. We added this information to provide important context in L: 152-154.

Line 146: Unclear what the difference between the two ASV numbers is. Were there only 5,116 total ASV's identified among all soils?

In total, 5,116 unique ASVs were detected as growing (APE 180 > 5 %) across all experiments as described in L: 516 in the methods section. When filtering for ASVs that consistently grew in at least two replicates, we were left with 3,553 ASVs. We added this information to the numbers given in parentheses in L: 180.

Lines 150: Again, I think it would be more informative to discuss effect sizes; by how much did drought decrease relative growth rates?

We agree that including effect sizes is important. Whenever shifts in relative growth rates of shared taxa (Fig. 2 B) were significant, we discussed percentage shifts in relative growth rates caused by drought L: 190.

Line 164: Unclear why normality and homoscedasticity assumptions need to be met for a simple comparison of mean # of growing phyla in two treatments.

We tested for significant shifts in the richness of growing ASVs due to drought, future climate, or their interaction using two-way ANOVA as stated in L: 544. To not violate the requirements for ANOVA (normality and homoscedasticity), we only included phyla that fulfilled normality and homoscedasticity assumptions. We clarified this in L: 208.

Line 198: "While ASVs classified as these putative taxa..."

The section to which this comment refers was extended and re-written in more detail. We made sure to include the reviewer's comment in the new version.

Line 210: "...why results are often ambiguous..."

The suggested change was adopted. L: 271

Lines 241-242: Not clear from the discussion what evidence points to either horizontal and/or vertical gene transfer, unless the authors mean to say that no support for either mechanism was directly supported.

We thank the reviewer for pointing out that this section was unclear. Our data indicates drought tolerance (defined as taxa that were still able to grow under drought conditions L: 65,145) but does not provide evidence for the underlying physiological mechanisms including horizontal and/or vertical gene transfer. However, we detected drought tolerance in taxa of microbial groups generally not considered drought-enduring, such as the Proteobacteria. Hence, we speculated that certain traits related to drought tolerance might not be restricted to distinct phylogenetic groups such as the Actinobacteriota but are more widespread, possibly due to independent evolution or horizontal gene transfer. We rephrased the sentence to make this clearer. See L: 304-307.

Lines 311-320: Were all treatments (ambient, ambient + drought, future, future + drought) all subjected to the same 6-week rainout condition? Additionally, it's unclear whether 'drought' in the naming of treatments is referring to this short-term rainout treatment or the longer 6-year (since 2014) manipulation.

Only the "ambient + drought" and the "future climate + drought treatments" were subjected to the simulated summer drought (6 weeks). Thus, 'drought' refers to the simulated drought event. Future climate were manipulated since 2014. Hence, we are comparing the effects of drought in a current as well as in a simulated future climate including a 6-year-long pre-exposure to these conditions. In order to avoid misunderstandings, we clarified this in L: 398-400.

Line 344: Unclear which treatments 'respectively' received which ^{18}O enrichment levels.

The volumetric water contents of our soils were $31.6 \pm 1.8\%$ under ambient precipitation and $6.9 \pm 1.9\%$ under drought. Soil samples under ambient precipitation were incubated with 95 atom % ^{18}O and drought-affected samples were 75 atom % ^{18}O labeled water to reach a target soil water enrichment of 70% atom% ^{18}O . We clarified this in L: 425-428.

Line 357: Supplemental Fig. 8 (not 5). How does the speed with which convergence on an

average 18O enrichment of soil water affect results? For the future climate + drought treatment, an average was reached within 50 hours but for the future climate treatment, 18O enrichment of soil water was continuing to equilibrate (according to the model showing in Supp Fig. 8) up until through 100 hours of incubation.

Based on the equilibration curves and the speed of equilibration, we calculated the mean 18O soil water enrichment over the course of the incubation for each treatment, taking equilibration speed variations into account. These values were then used to correct for differences in equilibration speed (L: 518) when calculating relative growth rates. We added this information and changed the sentence structure to clarify in L: 442-443.

Line 370: During fractionation?

The suggested correction was adopted. L: 454.

Line 375: Fractions (not factions)

The suggested correction was adopted. L: 461.

Fig. 2: (Panel C) Unclear how the relative growth rates are calculated such that all values are less than 1. From Line 342, is it to be assumed that these are percentages of maximum possible growth, where maximum possible growth would be an APE of taxon matches that of soil water APE?

Relative growth rates (RGR) were calculated using the following equation representing growth per day and assuming linear growth (L: 518)

*$RGR = APE\ 18O\ taxon / (Average\ APE\ 18O\ soil\ water * days\ of\ incubation)$*

We thank the reviewer for addressing this point. In qSIP literature ^{18}O atom fraction excess (AFE), the decimal of ^{18}O APE, is often used and the unit of qSIP calculations (<https://journals.asm.org/doi/10.1128/AEM.02280-15>) using publicly available code. However, since ^{18}O APE is a more commonly used term, we decided to work with it instead of ^{18}O AFE for this manuscript. For relative growth rate calculations, we worked with the decimal of ^{18}O APE though, which ranges between 0-1. We choose this approach to make relative growth rates more comparable between studies using qSIP. Dividing a value ranging between 0-1 by another value ranging between 0-1 (decimal of average APE ^{18}O soil water), hence, resulted in values below one. We clarified this in the current version of the manuscript. See L: 518-520.

Reviewer #3 (Remarks to the Author):

General comments: This is important research and the method used is novel. Understanding the response of soil microbes to multiple interacting global change factors is critical for predicting whether future ecosystems will be a source or sink for atmospheric carbon and is also critical for managing ecosystems to maintain soil fertility (i.e., maintaining healthy levels of soil organic matter). The method itself is important because it allows for measuring microbial growth in dry soils (i.e., under drought conditions). While the results appear sound and are exciting, the authors miss an important opportunity early in the paper to set up a strong rationale for their work. The writing is often vague and skips around such that there isn't a clear, linear storyline. The authors also overlook (or fail to appropriately highlight) important previous work on microbial responses to soil drying/drought. This paper is certainly not the first to look at this topic. What is new and exciting is that the authors were able to determine "Who" is active and under what conditions. The paper has potential to make a significant contribution to the literature but needs to be reframed to meet that potential. I provide some specific comments below to assist the authors in revising the paper.

We thank the reviewer for their positive feedback as well as their criticisms. By re-writing parts of the abstract and introduction, we removed sections of vague language, tried to build a stronger storyline, and referenced important previous work. We hope that this improved the readability of the revised version of the manuscript as well as its accessibility to a larger audience.

Specific comments:

L41-42. Ending of sentence ("...alleviating the loss of growing taxa within distinct communities.") is unclear/vague. I realize that space is limited in the abstract but providing a more specific result would make the abstract clearer and more compelling. The last two sentences of the abstract are also quite general, and these could be combined into a single sentence to give more space for a specific result in the sentence above, while providing for a more concise, harder-hitting ending to the abstract.

We thank the reviewer for this valuable input and incorporated the comments. We provided more results with specific numbers to be less vague and reduced the broader statements at the end. See L: 37-44.

L44. Why the specific focus on agriculture here. Certainly having drought resistant taxa in any ecosystem will become increasingly important for maintaining critical ecosystem functions.

We removed this sentence in the revised manuscript version. See L: 44-45.

L45. Awkward sentence construction: "...predicting future drought effects needs drought experiments...".

We agree with the reviewer and as per the previous comment, removed this sentence in the revised manuscript version. See L: 44-45.

Introduction: there is little information that puts soil C and its stabilization/loss into context. While I understand that context as a soil microbial ecologist, I think you need to provide that for the general readership of Nature Communications. That is, why care about soil C? How might it be impacted by global change? The same goes for drought. Can you provide some more specific information about the potential frequency of drought under climate change? All that to say, the introductory paragraphs are a little simplistic and vague. You're missing an opportunity to build a strong rationale for your work.

We thank the reviewer for this important comment and addressed it in the first paragraph of the introduction. We provided more information about droughts as well as the importance of soils and soil carbon in the context of climate change and agriculture in L: 49-58.

L60-62. This is an interesting statistic, but it seems a given that most respiration would come from active microbes, and for a general audience who likely doesn't know how many

microbes (active or dormant) are in soils, this info isn't that informative without more context.

We thank the reviewer for this comment and provided context about the number of microbial cells in soil in L: 69-71.

L80-81. Is the method only possible for bacteria and archaea? Why weren't fungi also evaluated? Are there limitations to the method in this regard?

While it is possible to perform qSIP on fungi (<https://www.nature.com/articles/s41396-021-01114-6>) this has mostly been done in aquatic ecosystems and rarely in soils. There remain several methodological limitations with fungal qSIP such as ITS amplicon sequencing and the associated issues of accurate fungal taxonomic assignment. While some groups are working on it, fungal qSIP was beyond the scope of this study.

L93. Define "pre-adaptation". Are you using this in an evolutionary context? If not, I suggest using a different term.

Since we did not mean to use "pre-adaptation" in an evolutionary context, we used the term "pre-exposure" instead. See L: 111.

L102. I don't think "Non-metric Multidimensional Scaling" should be capitalized.

Based on reviewer's comments, we performed PCA instead of NMDS.

L105-108. Very neat and important result!

We thank the reviewer for this positive comment which is also an important aspect of giving feedback.

L206. Consider rewording this sentence. There's been extensive work on microbial responses to drought/drying (e.g., significant work on this topic area by Mary Firestone and Josh Schimel, among others), but what your study provides is a look at who's active and under

what conditions. I think you need to do a better job both here and in the introduction of acknowledging previous work, while explaining the novelty of your own.

Our work was indeed based on and inspired by previous work from Josh Schimel and Mary Firestone, amongst others. We thank the reviewer for pointing out that this did not come through here and in the introduction. We re-worded the sentence in L: 266-268 as well as re-structured and partly re-wrote the introduction while referencing important studies.

L305. What are the dominant plant species in the system? Do they vary across the experimental treatments? What is the soil type and general soil characteristics (i.e., texture, pH, C&N content)? This manuscript should include some basic information and not require the reader to go to another paper to find.

We agree that including this information is important and added a short description of soil characteristics and dominant plant species in L: 388-394. Based on unpublished plant species data from 2019 (see figure below), we found that future climate conditions but not drought had a significant effect on plants species composition (Two-way PERMANOVA; Drought: $F = 1.41$, $df = 1$, $R^2 = 0.069$, $p = 0.23$; Climate: $F = 5.7$, $df = 1$, $R^2 = 0.27$, $p = 0.001$, Climate x Drought: $F = 2.03$, $df = 1$, $R^2 = 0.09$, $p = 0.076$)

L324. Why 95°C when 105°C is standard for mineral soils?

We checked our notes again and samples were indeed dried at 105 °C. We corrected this in the text. See L: 406.

L334-335. This seems like a very small sample size. Are larger samples not possible with this method? Did you evaluate the effect of sample size on outcome?

Larger soil sample sizes with qSIP are possible and have been used before (e.g., 2 g). This is primarily useful when working with low biomass samples since several µg of DNA are often needed for ultracentrifugation. Using the water-vapor equilibration method, even 400 mg of soil have been successfully used (<https://onlinelibrary.wiley.com/doi/full/10.1111/qcb.15168>) which we extended to 500 mg, a standard amount of soil used for many available DNA extraction kits. Since we were interested in the microbial community and the used amounts of soil were sufficient to adequately study it, we did not test other sample sizes. Though, we do acknowledge that this is important for future studies, especially when working with low-biomass samples. Theoretically, there are no concerns with using the water vapor equilibration method on larger soil samples if the size of the vial and, thus, the equilibration curve calculations are adjusted.

L360. By “snap-frozen”, I assume you mean “flash-frozen”?

Yes, we meant flash-frozen. We corrected it in the text. See L: 444.

L375. “Fractions” not “factions”.

We corrected this mistake in the text. See L: 459.

REVIEWERS' COMMENTS

Reviewer #1 (Remarks to the Author):

I have read the response to reviewers and the revised manuscript. All of my previous comments and concerns have been addressed. I think this manuscript will make an excellent contribution.

Eric Morrison, PhD

Reviewer #2 (Remarks to the Author):

I believe the authors have adequately responded to my concerns, including a stronger description of the experimental design and interpretation of the vap-qSIP results. I have no further comments.

Reviewer #3 (Remarks to the Author):

[No comments for authors]